# UNKNOWN DOMAIN INCONSISTENCY MINIMIZATION FOR DOMAIN GENERALIZATION

**Seungjae Shin**[*1], **HeeSun Bae**[*1], **Byeonghu Na**[1], **Yoon-Yeong Kim**[2] **& Il-Chul Moon**[1,3]

[1]Department of Industrial and Systems Engineering, KAIST
[2]Department of Statistics, University of Seoul, [3]summary.ai
{tmdwo0910,cat2507,wp03052,icmoon}@kaist.ac.kr, yykim@uos.ac.kr

## ABSTRACT

The objective of domain generalization (DG) is to enhance the transferability of the model learned from a source domain to unobserved domains. To prevent over-fitting to a specific domain, Sharpness-Aware Minimization (SAM) reduces source domain's loss sharpness. Although SAM variants have delivered significant improvements in DG, we highlight that there's still potential for improvement in generalizing to unknown domains through the exploration on data space. This paper introduces an objective rooted in both parameter and data perturbed regions for domain generalization, coined Unknown Domain Inconsistency Minimization (UDIM). UDIM reduces the loss landscape inconsistency between source domain and unknown domains. As unknown domains are inaccessible, these domains are empirically crafted by perturbing instances from the source domain dataset. In particular, by aligning the loss landscape acquired in the source domain to the loss landscape of perturbed domains, we expect to achieve generalization grounded on these flat minima for the unknown domains. Theoretically, we validate that merging SAM optimization with the UDIM objective establishes an upper bound for the true objective of the DG task. In an empirical aspect, UDIM consistently out-performs SAM variants across multiple DG benchmark datasets. Notably, UDIM shows statistically significant improvements in scenarios with more restrictive domain information, underscoring UDIM's generalization capability in unseen domains. Our code is available at https://github.com/SJShin-AI/UDIM.

## 1 INTRODUCTION

Domain Generalization (DG) (Zhou et al., 2022; Wang et al., 2022) focuses on *domain shift* that arises when training and testing occur across distinct domains, i.e. a domain of real pictures in training, and a separate domain of cartoon images in testing. The objective of DG is to train a model on a given source domain dataset, and generalizes well on other unobserved domains. To address the domain discrepancy between the source domain and other domains, various methods have been proposed: 1) alignment-based methods (Li et al., 2021a; Wald et al., 2021); 2) augmentation-based methods (Qiao et al., 2020a; Zhou et al., 2021); and 3) regularization-based methods (Arjovsky et al., 2019; Krueger et al., 2021; Rame et al., 2022). While these methodologies have demonstrated promising results, they often underperform in settings where given domain information is particularly limited (Wang et al., 2021b; Qiao et al., 2020b). Also, most methods lack theoretical guarantees on the minimization of the target risk at the distribution level.

In contrast to the aforementioned methods, Sharpness-aware optimizers, which flatten the loss landscape over a perturbed parameter region, demonstrate promising performances in DG tasks (Zhang et al., 2023b; Wang et al., 2023). By optimizing the perturbed local parameter regions, these approaches relieve the model overfitting to a specific domain, thereby enhancing the adaptability of the model across various domains. Also, this concept has a solid theoretical foundation based on the parameter space analysis with PAC-Bayes theories (McAllester, 1999; Dziugaite & Roy, 2017).

While the perturbation methods based on the parameter space have shown promising improvements in DG tasks, this paper theoretically claims that perturbation rooted in the data space is essential for

---

[*]Equal contribution

robust generalization to unobserved domains. Accordingly, this paper introduces an objective that leverages both parameter and data perturbed regions for domain generalization. In implementation, our objective minimizes the loss landscape discrepancy between a source domain and unknown domains, where unknown domains are emulated by perturbing instances from the source domain datasets. Recognizing the loss landscape discrepancy as an *Inconsistency score* across different domains, we name our objective as Unknown Domain Inconsistency Minimization (UDIM).

Introduction of UDIM on the DG framework has two contributions. First, we theoretically prove that the integration of sharpness-aware optimization and UDIM objective becomes the upper bound of population risk for all feasible domains, without introducing unoptimizable terms. Second, we reformulate the UDIM objective into a practically implementable term. This is accomplished from deriving the worst-case perturbations for both parameter space and data space, each in a closed-form expression. Our experiments on various DG benchmark datasets illustrate that UDIM consistently improves the generalization ability of parameter-region based methods. Moreover, we found that these improvements become more significant as domain information becomes more limited.

## 2 PRELIMINARY

### 2.1 PROBLEM DEFINITION OF DOMAIN GENERALIZATION

This paper investigates the task of domain generalization in the context of multi-class classification (Arjovsky et al., 2019; Sagawa et al., 2019; Nam et al., 2021). We define an input as $x \in \mathbb{R}^d$ and its associated class label as $y \in \{1, .., C\}$. Let $\mathscr{D}_e$ represent a distribution of $e$-th domain. $\mathcal{E}$ is a set of indices for all domains, and $\mathscr{D} := \{\mathscr{D}_e\}_{e \in \mathcal{E}}$ denotes the set of distributions for all domains, where every domain shares the same class set. For instance, let's hypothesize that video streams from autonomous cars are being collected, and the data collection at days and nights will constitute two distinct domains. A sampled dataset from the $e$-th domain is denoted by $D_e = \{(x_i, y_i)\}_{i=1}^{n_e}$ where $(x_i, y_i) \sim \mathscr{D}_e$ and $n_e$ is the number of data instances of $e$-th domain.

Throughout this paper, let $\theta \in \Theta$ represents a parameter of trained model $f_\theta$, where $\Theta$ is a set of model parameters. Using $D_e$, we define a loss function as $\mathcal{L}_{D_e}(\theta) = \frac{1}{n_e} \sum_{(x_i, y_i) \in D_e} \ell(f_\theta(x_i), y_i)$, where we sometimes denote $\ell(f_\theta(x_i), y_i)$ as $\ell(x_i, \theta)$. The population risk for domain $e$ is given by $\mathcal{L}_{\mathscr{D}_e}(\theta) = \mathbb{E}_{(x,y) \sim \mathscr{D}_e}[\ell(f_\theta(x), y)]$. Then, the population risk over all domains is defined as $\mathcal{L}_{\mathscr{D}}(\theta) = \sum_{e \in \mathcal{E}} p(e) \mathcal{L}_{\mathscr{D}_e}(\theta)$, where $p(e)$ represents the occurrence probability of domain $e$.

In essence, the primary goal of training a model, $f_\theta$, is to minimize the population risk, $\mathcal{L}_{\mathscr{D}}(\theta)$. In practical scenarios, we only have access to datasets derived from a subset of all domains. We call these accessible domains and datasets as *source domains* and *source domain datasets*, respectively; denoted as $\mathscr{D}_S = \{\mathscr{D}_s\}_{s \in \mathcal{S}}$ and $D_S = \{D_s\}_{s \in \mathcal{S}}$ where $\mathcal{S}$ is the set of indexes for source domains. As $\mathscr{D}_S \neq \mathscr{D}$, $D_S$ deviates from the distribution $\mathscr{D}$ under the sampling bias of $\mathscr{D}_S$. As a consequence, a model parameter $\theta_S^* = \mathrm{argmin}_\theta \mathcal{L}_{D_S}(\theta)$, which is trained exclusively on $D_S$, might not be optimal for $\mathcal{L}_{\mathscr{D}}(\theta)$. Accordingly, domain generalization emerges as a pivotal task to optimize $\theta^* = \mathrm{argmin}_\theta \mathcal{L}_{\mathscr{D}}(\theta)$ by only utilizing the source domain dataset, $D_S$.

### 2.2 VARIANTS OF SHARPNESS-AWARE MINIMIZATION FOR DOMAIN GENERALIZATION

Recently, a new research area has emerged by considering optimization over the parameter space (Foret et al., 2020; Kwon et al., 2021). Several studies have focused on the problem of $\theta$ overfitted to a training dataset (Wang et al., 2023; Zhang et al., 2023b;a). These studies confirmed that optimization on the perturbed parameter region improves the generalization performance of the model. To construct a model that can adapt to unknown domains, it is imperative that an optimized parameter point is not overfitted to the source domain datasets. Accordingly, there were some studies to utilize the parameter perturbation in order to avoid such overfitting, as elaborated below (Table 1).

Among variations in Table 1, Sharpness-Aware Minimization (SAM) (Foret et al., 2020) is the most basic form of the parameter perturbation, which regularizes the local region of $\theta$ to be the flat minima on the loss curvature as $\min_\theta \max_{\|\epsilon\|_2 \leq \rho} \mathcal{L}_{D_s}(\theta + \epsilon) + \|\theta\|_2$. Here, $\epsilon$ is the perturbation vector to $\theta$; and $\rho$ is the maximum size of the perturbation vector. Subsequently, methodologies for regularizing stronger sharpness were introduced (Zhang et al., 2023b; Wang et al., 2023; Zhang et al., 2023a).

These approaches exhibited incremental improvements in domain generalization tasks. Check Appendix A for more explanations.

Table 1: Objectives of SAM variants for DG

| Method | Objective |
|---|---|
| SAM (Foret et al., 2020) | $\max_{\|\epsilon\|_2 \leq \rho} \mathcal{L}_{D_s}(\theta + \epsilon)$ |
| GAM (Zhang et al., 2023b) | $\mathcal{L}_{D_s}(\theta) + \rho \max_{\|\epsilon\|_2 \leq \rho} \|\nabla \mathcal{L}_{D_s}(\theta + \epsilon)\|$ |
| SAGM (Wang et al., 2023) | $\mathcal{L}_{D_s}(\theta) + \mathcal{L}_{D_s}(\theta + \rho \nabla \mathcal{L}_{D_s}(\theta) / \|\nabla \mathcal{L}_{D_s}(\theta)\| - \alpha \nabla \mathcal{L}_{D_s}(\theta))$ |
| FAM (Zhang et al., 2023a) | $\mathcal{L}_{D_s}(\theta) + \max_{\|\epsilon\|_2 \leq \rho} (\mathcal{L}_{D_s}(\theta + \epsilon) - \mathcal{L}_{D_s}(\theta)) + \rho \max_{\|\epsilon\|_2 \leq \rho} \|\nabla \mathcal{L}_{D_s}(\theta + \epsilon)\|$ |

We are motivated by this characteristic to extend the parameter perturbation towards data perturbation, which could be effective for the exploration of unknown domains. None of SAM variants proposed data-based perturbation for unknown domain discovery. Fundamentally, SAM variants predominantly focus on identifying the flat minima for the given source domain dataset, $D_S$. In Section 3.1, we highlight that finding flat minima in the source domain cannot theoretically ensure generalization to unknown target domains. Consequently, we demonstrate that generalization should be obtained from unobserved domains, rather than solely the source domain. Since SAM imposes the perturbation radius of $\rho$ on only $\theta$, we hypothesize $D_s$ could be perturbed by additional mechanism to generalize $D_s$ toward $\mathscr{D}$.

While SAM minimizes the loss over $\rho$-ball region of $\theta$, its actual implementation minimizes the maximally perturbed loss w.r.t. $\theta + \epsilon^*$; where the maximal perturbation, $\epsilon^*$, is approximated in a closed-form solution via Taylor expansion as follows:

$$\max_{\|\epsilon\|_2 \leq \rho} \mathcal{L}_{D_s}(\theta + \epsilon) \approx \mathcal{L}_{D_s}(\theta + \epsilon^*) \text{ where } \epsilon^* \approx \rho \cdot \text{sign}(\nabla_\theta \mathcal{L}_{D_s}(\theta)) \frac{|\nabla_\theta \mathcal{L}_{D_s}(\theta)|}{\|\nabla_\theta \mathcal{L}_{D_s}(\theta)\|_2^2}. \tag{1}$$

The existence of this closed-form solution of $\epsilon^*$ simplifies the learning procedure of SAM by avoiding min-max game and its subsequent problems, such as oscillation of parameters (Chu et al., 2019). This closed-form solution can also be applied to the perturbation on the data space.

## 3 METHOD

### 3.1 MOTIVATION : BEYOND THE SOURCE DOMAIN-BASED FLATNESS

Based on the context of domain generalization, Theorem 3.1 derives the relationship between the SAM loss on the source domain dataset, denoted as $\max_{\|\epsilon\|_2 \leq \rho} \mathcal{L}_{D_s}(\theta + \epsilon)$, and the generalization loss on an arbitrary unknown domain, $\mathcal{L}_{\mathscr{D}_e}(\theta)$, for a model parameter $\theta \in \Theta$ as follows:

**Theorem 3.1.** *(Rangwani et al., 2022) For $\theta \in \Theta$ and arbitrary domain $\mathscr{D}_e \in \mathscr{D}$, with probability at least $1 - \delta$ over realized dataset $D_s$ from $\mathscr{D}_s$ with $|D_s| = n$, the following holds under some technical conditions on $\mathcal{L}_{\mathscr{D}_e}(\theta)$, where $h_0 : \mathbb{R}_+ \to \mathbb{R}_+$ is a strictly increasing function.*

$$\mathcal{L}_{\mathscr{D}_e}(\theta) \leq \max_{\|\epsilon\|_2 \leq \rho} \mathcal{L}_{D_s}(\theta + \epsilon) + \mathcal{D}^f(\mathscr{D}_s || \mathscr{D}_e) + h_0(\frac{\|\theta\|_2^2}{\rho^2}) \tag{2}$$

Theorem 3.1 provides an upper bound for $\mathcal{L}_{\mathscr{D}_e}(\theta)$. The second term of this upper bound, represented as $\mathcal{D}^f(\mathscr{D}_s || \mathscr{D}_e)$, corresponds to the distribution discrepancy between $\mathscr{D}_s$ and $\mathscr{D}_e$. Notably, $\mathcal{D}^f$ denotes the discrepancy based on $f$-divergence. When $e \neq s$, $D_e$ is inaccessible in the context of domain generalization. Accordingly, the SAM optimization on $D_s$ leaves an unoptimizable term in its upper bound, posing challenges for the domain generalization.

In a setting that only $D_s$ and model parameter $\theta$ are accessible, generating unseen domain data becomes infeasible. Nonetheless, by perturbing $D_s$ towards the direction that is most sensitive given $\theta$, we can emulate the worst-case scenario for an unobserved domain (Sagawa et al., 2019). While parameters trained via the SAM optimizer may exhibit flat regions based on the source domain dataset, $D_s$, there is no theoretic study on the flat minima under unobserved domains. By identifying the worst-case scenario that maximizes the loss landscape difference between domains, our methodology seeks the generalization across the unknown domains.

### 3.2 UDIM : UNKNOWN DOMAIN INCONSISTENCY MINIMIZATION

This section proposes an objective based on both parameter and data perturbed regions for domain generalization, coined **U**nknown **D**omain **I**nconsistency **M**inimization (UDIM). UDIM minimizes

(a) Changes in the loss landscape across domains based on the perturbed parameter space of $\theta$: initial state (left), post-SAM optimization (center), and subsequent to UDIM application (right).

(b) Changes in domain-wise inconsistency sharpness based on data space of $D_s$ before (left) and after (right) applying UDIM.

Figure 1: Illustration of our model, UDIM, based on parameter space (a) and data space (b). (a) We define flatness within a perturbed region by minimizing the inconsistency loss relative to the unknown domains, around the flat region derived from the source domain. (b) Furthermore, by reducing the domain-wise inconsistency within the input perturbed regions, where $\rho_x$ denotes perturbation length, our method can also be interpreted as an data space perspective of SAM.

the loss landscape discrepancy between the source domain and unknown domains, where unknown domains are empirically realized by perturbing instances from the source domain dataset, $D_s$.

Let $\Theta_{\theta,\rho} = \{\theta' | \|\theta' - \theta\|_2 \leq \rho\}$, which is $\rho$-ball perturbed region of a specific parameter point $\theta$. When training a parameter $\theta$ on an arbitrary domain dataset $D_e$ with the SAM optimizer, some regions within $\Theta_{\theta,\rho}$ are expected to be optimized as flat regions for source domain. Following the notation of Parascandolo et al. (2020b), define $N_{e,\theta}^{\gamma,\rho} := \{\theta' \in \Theta_{\theta,\rho} \,|\, |\mathcal{L}_{D_e}(\theta') - \mathcal{L}_{D_e}(\theta)| \leq \gamma\}$, which is the region in $\Theta_{\theta,\rho}$ where the loss value of $\theta$ deviates by no more than $\gamma$. Given small enough values of $\mathcal{L}_{D_e}(\theta)$ and $\gamma$, $N_{e,\theta}^{\gamma,\rho}$ could be recognized as a flat minima for $e$-th domain.

We aim to utilize the flat minima of $\theta$ obtained through training on $D_s$ using the SAM optimizer, where we represent $s$-th domain as our source domain. The goal is to regularize the loss and its corresponding landscape of unknown domains, so that the flat minima of the unknown domain aligns with that of the source domain. By optimizing the domain of worst-case deviation in the loss landscape from $D_s$, we facilitate regularization across a range of intermediary domains. Eq. 3 formalizes our motivation, which is *cross-domain inconsistency score*:

$$\mathcal{I}_s^\gamma(\theta) = \max_{e \in \mathcal{E}} \max_{\theta' \in N_{s,\theta}^{\gamma,\rho}} |\mathcal{L}_{D_e}(\theta') - \mathcal{L}_{D_s}(\theta')| \qquad (3)$$

In the above equation, the inner maximization seeks the worst-case parameter point $\theta'$ that amplifies the domain-wise loss disparity, while the outer maximization identifies $e$-th domain that maximizes $\max_{\theta' \in N_{s,\theta}^{\gamma,\rho}} |\mathcal{L}_{D_e}(\theta') - \mathcal{L}_{D_s}(\theta')|$. Parascandolo et al. (2020a) utilizes an equation similar to Eq. 3, and this equation is also employed to indicate $\theta$ with a loss surface that is invariant to the environment changes. Our methodology differs from Parascandolo et al. (2020a), which simply uses inconsistency as a metric, in that we employ it directly as the objective we aim to optimize. Let's assume that $\theta$ exhibits sufficiently low loss along with flat loss landscape of $D_s$. Further, if we can identify $\theta$ with a low value of $\mathcal{I}_s^\gamma(\theta)$, then this $\theta$ would also demonstrate consistently good generalization for unknown domains. This motivation leads to the UDIM's objective, which specifically targets the reduction of the cross-domain inconsistency score across unobserved domains.

**Objective Formulation of UDIM**   While we define the cross-domain inconsistency as $\mathcal{I}_s^\gamma(\theta)$, the final formulation of UDIM is the optimization of $\theta$ regarding both source and unknown domain losses from the flat-minima perspective. Eq. 4 is the parameter optimization of our proposal:

$$\min_\theta \left( \max_{\|\epsilon\|_2 \leq \rho} \mathcal{L}_{D_s}(\theta + \epsilon) + \lambda_1 \max_{e \in \mathcal{E}} \max_{\theta' \in N_{s,\theta}^{\gamma,\rho}} |\mathcal{L}_{D_e}(\theta') - \mathcal{L}_{D_s}(\theta')| + \lambda_2 \|\theta\|_2 \right) \qquad (4)$$

The objective of UDIM consists of three components. The first term is a SAM loss on $D_s$, guiding $\theta$ toward a flat minima in the context of $D_s$. Concurrently, as $\theta$ provides flattened local landscape, the second term weighted with $\lambda_1$ is a region-based loss disparity between the worst-case domain $e$ and the source domain. Subsequently, the algorithm seeks to diminish the region-based loss disparity between the worst-case domain and the source domain. Figure 1 (a) illustrates the update of the loss landscape in the parameter-space for each domain, based on the optimization of Eq. 4. As SAM

optimization on $D_s$ progresses, $N_{s,\theta}^{\gamma,\rho}$ is expected to be broadened. However, SAM optimization does not imply the minimization of $\mathcal{I}_s^\gamma(\theta)$ (center). Given that the optimization of $\mathcal{I}_s^\gamma(\theta)$ is conducted on $N_{s,\theta}^{\gamma,\rho}$, we expect the formation of flat minima spanning all domains in the optimal state (rightmost).

The optimization of Eq. 4 can also be interpreted as minimizing another form of sharpness in the data space. Let's suppose all other domains lie within a finite perturbation of the source domain's dataset. In the context of the data space over the $D_s$, the optimization can be seen as identifying the worst-case perturbed dataset, $D_e$, by evaluating $\max_{\theta' \in N_{s,\theta}^{\gamma,\rho}} |\mathcal{L}_{D_e}(\theta') - \mathcal{L}_{D_s}(\theta')|$. If we view the perturbation of the $D_s$, not as discrete choices but as a continuum within the data-space; our optimization can be viewed as minimizing the sharpness of domain-wise inconsistency in the data space. Figure 1 (b) illustrates this interpretation. After such optimization, the resulting parameter $\theta$ can offer the consistent generalization over domains located within the perturbed data space.

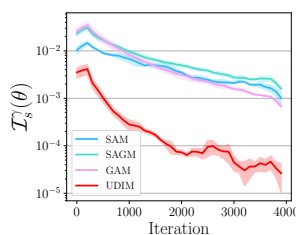

Figure 2: Inconsistency score of each method on PACS training dataset (X-axis: training iteration). Y-axis is depicted in a log-scale.

At this juncture, a crucial consideration is whether the SAM optimization on $D_s$ focuses on minimizing the second term, $\mathcal{I}_s^\gamma(\theta)$, or not. We illustrate the limited capability of SAM variants in minimizing the second term, by an analytical illustration of Figure 1 (a); and by an empirical demonstration of Figure 2. Therefore, UDIM covers the optimization area that SAM does not operate.

**Theoretical Analysis of UDIM**   Given the definition of $\Theta_{\theta,\rho} = \{\theta' | \|\theta' - \theta\|_2 \leq \rho\}$, we introduce $\Theta_{\theta,\rho'} = \arg\max_{\Theta_{\theta,\hat\rho} \subseteq N_{s,\theta}^{\gamma,\rho}} \hat\rho$, which is the largest $\rho'$-ball region around $\theta$ in $N_{s,\theta}^{\gamma,\rho}$.[1] Theorem 3.2 introduces the generalization bound of Eq. 4, which is the objective of UDIM. Theorem 3.2 states that Eq. 4 can become the upper bound of $\mathcal{L}_{\mathscr{D}}(\theta)$, which is the population risk over all domains.

**Theorem 3.2.** *For $\theta \in \Theta$ and arbitrary domain $e \in \mathcal{E}$, with probability at least $1 - \delta$ over realized dataset $D_e$ from $\mathscr{D}_e$, the following holds under technical conditions on $\mathcal{L}_{\mathscr{D}_e}(\theta)$ and $\mathcal{L}_{D_e}(\theta)$, where $h : \mathbb{R}_+ \to \mathbb{R}_+$ is a strictly increasing function. (Proof in Appendix B.1.)*

$$\mathcal{L}_{\mathscr{D}}(\theta) \leq \max_{\|\epsilon\|_2 \leq \rho} \mathcal{L}_{D_s}(\theta + \epsilon) + (1 - \frac{1}{|\mathcal{E}|}) \max_{e \in \mathcal{E}} \max_{\theta' \in N_{s,\theta}^{\gamma,\rho}} |\mathcal{L}_{D_e}(\theta') - \mathcal{L}_{D_s}(\theta')| + h(\frac{\|\theta\|_2^2}{\rho^2}) \quad (5)$$

In such a case, Theorem 3.2 retains the same form of weight decay term as presented in Theorem 3.1. Unlike Theorem 3.1, which contains terms that are inherently unoptimizable, our objective is capable of minimizing every term encompassed in the upper bound of Theorem 3.2 because we do not have an inaccessible term, $\mathcal{D}^f(\mathscr{D}_s || \mathscr{D}_e)$.

## 3.3   Implementation of UDIM

This section reformulates the second term of Eq. 4 into an implementable form. We first formalize the perturbation of $D_s$ to emulate the worst-case domain for $\max_{\theta' \in N_{s,\theta}^{\gamma,\rho}} |\mathcal{L}_{D_e}(\theta') - \mathcal{L}_{D_s}(\theta')|$.

**Inconsistency-Aware Domain Perturbation on $D_s$**   For clarification, we explain the perturbation process based on an arbitrary input instance $x$ from $D_s$. Given that the magnitude of the perturbation vector for input $x$ is constrained to $\rho_x$, the perturbed input $\tilde{x}$ can be expressed as follows:

$$\tilde{x} = x + \operatorname*{argmax}_{\epsilon_x : \|\epsilon_x\|_2 \leq \rho_x} \max_{\theta' \in N_{s,\theta}^{\gamma,\rho}} \left( \ell(x + \epsilon_x, \theta') - \ell(x, \theta') \right) \approx x + \operatorname*{argmax}_{\epsilon_x : \|\epsilon_x\|_2 \leq \rho_x} \max_{\theta' \in N_{s,\theta}^{\gamma,\rho}} \ell(x + \epsilon_x, \theta')$$

$$(6)$$

$$\underset{\text{1st Taylor}}{\approx} x + \operatorname*{argmax}_{\epsilon_x : \|\epsilon_x\|_2 \leq \rho_x} \left( \ell(x + \epsilon_x, \theta) + \rho' \|\nabla_\theta \ell(x + \epsilon_x, \theta)\|_2 \right) \quad (7)$$

Assuming $N_{s,\theta}^{\gamma,\rho}$ as flat minima of $\theta$, $\ell(x, \theta')$ is almost invariant for $\theta' \in N_{s,\theta}^{\gamma,\rho}$. We assume that the invariant value of $\ell(x, \theta')$ does not significantly influence the direction of $x$'s perturbation because it becomes almost constant in $N_{s,\theta}^{\gamma,\rho}$. Therefore, we cancel out this term in Eq. 6.

---

[1]For the sake of simplicity, we do not inject the notation of $s$ or $\gamma$ in $\Theta_{\theta,\rho'}$.

Additionally, as we cannot specify the regional shape of $N_{s,\theta}^{\gamma,\rho}$, it is infeasible to search maximal point $\theta' \in N_{s,\theta}^{\gamma,\rho}$. As a consequence, we utilize $\Theta_{\theta,\rho'}$, which is the largest $\rho'$-ball region within $N_{s,\theta}^{\gamma,\rho}$, to approximately search the maximal point $\theta'$. It should be noted that $\Theta_{\theta,\rho'} \subseteq N_{s,\theta}^{\gamma,\rho} \subseteq \Theta_{\theta,\rho}$, where we assume that $\rho'$ gradually approaches $\rho$ during the SAM optimization. Through the first-order Taylor expansion[2] for the maximum point within $\Theta_{\rho'}$, we can design the aforementioned perturbation loss as Eq. 7. Consequently, the perturbation is carried out in a direction that maximizes the loss and $\rho'$-weighted gradient norm of the original input $x$.

**Inconsistency Minimization on $\theta$**  After the perturbation on $x \in D_s$, we get an inconsistency-aware perturbed dataset, $\tilde{D}_s$; which approximates the worst-case of unobserved domains. Accordingly, we can formulate the optimization of $\mathcal{I}_s^\gamma(\theta)$ based on $\theta$ as $\min_\theta \max_{\theta' \in N_{s,\theta}^{\gamma,\rho}} \left( \mathcal{L}_{\tilde{D}_s}(\theta') - \mathcal{L}_{D_s}(\theta') \right)$.

For the above min-max optimization, we approximate the search for the maximum parameter $\theta'$ in a closed-form using second order taylor-expansion, similar to the approach of SAM in Eq. 1.

$$\max_{\theta' \in N_{s,\theta}^{\gamma,\rho}} \left( \mathcal{L}_{\tilde{D}_s}(\theta') - \mathcal{L}_{D_s}(\theta') \right) \approx \mathcal{L}_{\tilde{D}_s}(\theta) - \mathcal{L}_{D_s}(\theta) + \rho' \|\nabla_\theta \mathcal{L}_{\tilde{D}_s}(\theta)\|_2 + \max_{\theta' \in N_{s,\theta}^{\gamma,\rho}} \frac{1}{2} \theta'^\top \mathbf{H}_{\tilde{D}_s} \theta' \tag{8}$$

$$= \left( \mathcal{L}_{\tilde{D}_s}(\theta) - \mathcal{L}_{D_s}(\theta) \right) + \rho' \|\nabla_\theta \mathcal{L}_{\tilde{D}_s}(\theta)\|_2 + \gamma \max_i \lambda_i^{\tilde{D}_s} / \lambda_i^{D_s} \tag{9}$$

Full derivation and approximation procedure of Eq. 8 and Eq. 9 are in Appendix B.2. $\mathbf{H}_{\tilde{D}_s}$ in Eq. 8 denotes a Hessian matrix of the perturbed dataset, $\tilde{D}_s$. Also, $\lambda_i^{\tilde{D}_s}$ in Eq. 9 denotes $i$-th eigenvalue of $\mathbf{H}_{\tilde{D}_s}$. Finally, Eq. 9 becomes the objective with three components: 1) the loss difference between $\tilde{D}_s$ and $D_s$, 2) the gradient norm of $\tilde{D}_s$ and 3) the maximum eigenvalue ratio between $\tilde{D}_s$ and $D_s$. Note that $\lambda_i^{\tilde{D}_s} / \lambda_i^{D_s}$ is minimized when $\mathbf{H}_{\tilde{D}_s}$ becomes equivalent to $\mathbf{H}_{D_s}$.

While Eq. 9 is differentiable with respect to $\theta$ and can thus be utilized as a tractable objective, computing the Hessian matrix for an over-parameterized $\theta$ is computationally demanding. In line with Rame et al. (2022), we replace the objective with the Hessian matrix (Eq. 9) with an optimization based on gradient variance. Accordingly, Eq. 10 represents the gradient variance-based objective as an optimization with respect to $\theta$ as follows: (See detailed derivations in Appendix B.3)

$$\min_\theta \rho' \|\nabla_\theta \mathcal{L}_{\tilde{D}_s}(\theta)\|_2 + \|\text{Var}(\mathbf{G}_{\tilde{D}_s}) - \text{Var}(\mathbf{G}_{D_s})\|_2 \tag{10}$$

Here, $\mathbf{g}_i$ is a per-sample gradient for $i$-th sample; and $\mathbf{G}_D = \{\mathbf{g}_i\}_{i=1}^{|D|}$ is a set of per-sample gradient for $x_i \in D$. Accordingly, the variance of $\mathbf{G}_D$, which we denote as $\text{Var}(\mathbf{G}_D)$, is calculated as $\text{Var}(\mathbf{G}_D) = \frac{1}{|D|-1} \sum_{i=1}^{|D|} (\mathbf{g}_i - \bar{\mathbf{g}})^2$. Matching the gradient variances between two distinct datasets encapsulates a specific form of loss matching between them (Rame et al., 2022). This allows us to unify loss matching and Hessian matching under a single optimization using $\text{Var}(\mathbf{G}_D)$.

**Summary** Our methodology applies the perturbation technique to both input $x$ and parameter $\theta$. Particularly, the perturbation on inputs and parameters is necessary to minimize $\mathcal{I}_s^\gamma(\theta)$ under unknown domains, which is the unique contribution of this work. Ablation study in Section 4.3 supports that UDIM, which is the combination of the Eq. 7 and Eq. 10, yields the best performance compared to various perturbations and optimizations. Algorithm of UDIM is in Appendix C.

A single iteration of UDIM optimization with SAM loss can be described as the following procedure:

1. Construction of $\tilde{D}_s$ via Inconsistency-Aware Domain Perturbation on $D_s$

$$\tilde{D}_s = \{(\tilde{x}_i, y_i) \mid (x_i, y_i) \in D_s\} \text{ where } \tilde{x}_i = x_i + \rho_x \frac{\nabla_{x_i}\left(\ell(x,\theta_t) + \rho'\|\nabla_{\theta_t}\ell(x,\theta_t)\|_2\right)}{\left\|\nabla_{x_i}\left(\ell(x,\theta_t) + \rho'\|\nabla_{\theta_t}\ell(x,\theta_t)\|_2\right)\right\|_2} \tag{11}$$

2. SAM loss and Inconsistency Minimization on the current parameter $\theta_t$

$$\theta_{t+1} = \theta_t - \eta \nabla_{\theta_t} \left( \max_{\|\epsilon\|_2 \leq \rho} \mathcal{L}_{D_s}(\theta_t + \epsilon) + \rho'\|\nabla_{\theta_t}\mathcal{L}_{\tilde{D}_s}(\theta_t)\|_2 + \|\text{Var}(\mathbf{G}_{\tilde{D}_s}) - \text{Var}(\mathbf{G}_{D_s})\|_2 + \lambda_2\|\theta_t\|_2 \right) \tag{12}$$

---

[2] We empirically found out that extending it into second-order does not affect the resulting performance.

Table 2: Test accuracy for CIFAR-10-C. Each level states the severity of corruption. **Bold** is the best case of each column or improved performances with the respective sharpness-based optimizers.

| | Method | level1 | level2 | level3 | level4 | level5 | Avg |
|---|---|---|---|---|---|---|---|
| | ERM | $75.9_{\pm 0.5}$ | $72.9_{\pm 0.4}$ | $70.0_{\pm 0.4}$ | $65.9_{\pm 0.4}$ | $59.9_{\pm 0.5}$ | 68.9 |
| | IRM (Arjovsky et al., 2019) | $37.6_{\pm 2.7}$ | $36.0_{\pm 2.8}$ | $34.6_{\pm 2.6}$ | $32.8_{\pm 2.1}$ | $30.8_{\pm 1.9}$ | 34.3 |
| | GroupDRO (Sagawa et al., 2019) | $76.0_{\pm 0.1}$ | $72.9_{\pm 0.1}$ | $69.8_{\pm 0.2}$ | $65.5_{\pm 0.3}$ | $59.5_{\pm 0.5}$ | 68.7 |
| | OrgMixup (Zhang et al., 2018) | $77.1_{\pm 0.0}$ | $74.2_{\pm 0.1}$ | $71.4_{\pm 0.1}$ | $67.4_{\pm 0.2}$ | $61.2_{\pm 0.1}$ | 70.3 |
| | Mixup (Yan et al., 2020) | $76.3_{\pm 0.3}$ | $73.2_{\pm 0.2}$ | $70.2_{\pm 0.2}$ | $66.1_{\pm 0.1}$ | $60.1_{\pm 0.1}$ | 69.2 |
| | CutMix (Yun et al., 2019) | $77.9_{\pm 0.0}$ | $74.2_{\pm 0.1}$ | $70.8_{\pm 0.2}$ | $66.3_{\pm 0.3}$ | $60.0_{\pm 0.4}$ | 69.8 |
| | MTL (Blanchard et al., 2021) | $75.6_{\pm 0.5}$ | $72.7_{\pm 0.4}$ | $69.9_{\pm 0.2}$ | $65.9_{\pm 0.0}$ | $60.2_{\pm 0.3}$ | 68.9 |
| LOO | MMD (Li et al., 2018b) | $76.4_{\pm 0.1}$ | $73.2_{\pm 0.2}$ | $70.0_{\pm 0.3}$ | $65.7_{\pm 0.3}$ | $59.6_{\pm 0.5}$ | 69.0 |
| DG | CORAL Sun & Saenko (2016) | $76.0_{\pm 0.4}$ | $72.9_{\pm 0.2}$ | $69.9_{\pm 0.0}$ | $65.8_{\pm 0.1}$ | $59.6_{\pm 0.1}$ | 68.8 |
| Based | SagNet (Nam et al., 2021) | $76.6_{\pm 0.2}$ | $73.6_{\pm 0.3}$ | $70.5_{\pm 0.4}$ | $66.4_{\pm 0.4}$ | $60.1_{\pm 0.4}$ | 69.5 |
| | ARM (Zhang et al., 2021) | $75.7_{\pm 0.1}$ | $72.9_{\pm 0.1}$ | $69.9_{\pm 0.2}$ | $65.9_{\pm 0.2}$ | $59.8_{\pm 0.3}$ | 68.8 |
| | DANN (Ganin et al., 2016) | $75.4_{\pm 0.4}$ | $72.6_{\pm 0.3}$ | $69.7_{\pm 0.2}$ | $65.6_{\pm 0.0}$ | $59.6_{\pm 0.2}$ | 68.6 |
| | CDANN (Li et al., 2018c) | $75.3_{\pm 0.2}$ | $72.3_{\pm 0.2}$ | $69.4_{\pm 0.2}$ | $65.3_{\pm 0.1}$ | $59.4_{\pm 0.2}$ | 68.3 |
| | VREx Krueger et al. (2021) | $76.0_{\pm 0.2}$ | $73.0_{\pm 0.2}$ | $70.0_{\pm 0.2}$ | $66.0_{\pm 0.1}$ | $60.0_{\pm 0.2}$ | 69.0 |
| | RSC (Huang et al., 2020) | $76.1_{\pm 0.4}$ | $73.2_{\pm 0.5}$ | $70.1_{\pm 0.5}$ | $66.2_{\pm 0.5}$ | $60.1_{\pm 0.5}$ | 69.1 |
| | Fishr (Rame et al., 2022) | $76.3_{\pm 0.3}$ | $73.4_{\pm 0.3}$ | $70.4_{\pm 0.5}$ | $66.3_{\pm 0.8}$ | $60.1_{\pm 1.1}$ | 69.3 |
| SDG | M-ADA (Qiao et al., 2020a) | $77.2_{\pm 0.2}$ | $74.2_{\pm 0.1}$ | $71.2_{\pm 0.0}$ | $67.1_{\pm 0.1}$ | $61.1_{\pm 0.1}$ | 70.2 |
| Based | LTD (Wang et al., 2021a) | $75.3_{\pm 0.2}$ | $73.0_{\pm 0.0}$ | $70.6_{\pm 0.0}$ | $67.2_{\pm 0.2}$ | $61.7_{\pm 0.2}$ | 69.6 |
| | SAM (Foret et al., 2020) | $79.0_{\pm 0.3}$ | $76.0_{\pm 0.3}$ | $72.9_{\pm 0.3}$ | $68.7_{\pm 0.2}$ | $62.5_{\pm 0.3}$ | 71.8 |
| | **UDIM** w/ SAM | $\mathbf{80.3}_{\pm 0.0}$ | $\mathbf{77.7}_{\pm 0.1}$ | $\mathbf{75.1}_{\pm 0.0}$ | $\mathbf{71.5}_{\pm 0.1}$ | $\mathbf{66.2}_{\pm 0.1}$ | **74.2** |
| SAM | SAGM (Wang et al., 2023) | $79.0_{\pm 0.1}$ | $76.2_{\pm 0.0}$ | $73.2_{\pm 0.2}$ | $69.0_{\pm 0.3}$ | $62.7_{\pm 0.4}$ | 72.0 |
| Based | **UDIM** w/ SAGM | $\mathbf{80.1}_{\pm 0.1}$ | $\mathbf{77.5}_{\pm 0.1}$ | $\mathbf{74.8}_{\pm 0.1}$ | $\mathbf{71.2}_{\pm 0.2}$ | $\mathbf{65.9}_{\pm 0.2}$ | **73.9** |
| | GAM (Zhang et al., 2023b) | $79.5_{\pm 0.1}$ | $76.8_{\pm 0.1}$ | $74.0_{\pm 0.2}$ | $69.9_{\pm 0.1}$ | $64.1_{\pm 0.2}$ | 72.8 |
| | **UDIM** w/ GAM | $\mathbf{81.4}_{\pm 0.1}$ | $\mathbf{78.9}_{\pm 0.0}$ | $\mathbf{76.3}_{\pm 0.0}$ | $\mathbf{72.8}_{\pm 0.1}$ | $\mathbf{67.4}_{\pm 0.1}$ | **75.3** |

# 4 EXPERIMENT

## 4.1 IMPLEMENTATION

**Datasets and Implementation Details** We validate the efficacy of our method, UDIM, via experiments across multiple datasets for domain generalization. First, we conducted evaluation on CIFAR-10-C (Hendrycks & Dietterich, 2019), a synthetic dataset that emulates various domains by applying several synthetic corruptions to CIFAR-10 (Krizhevsky et al., 2009). Furthermore, we extend our evaluation to real-world datasets with multiple domains, namely PACS (Li et al., 2017), Office-Home (Venkateswara et al., 2017), and DomainNet (Peng et al., 2019). Since UDIM can operate regardless of the number of source domains, we evaluate UDIM under both scenarios on real-world datasets: 1) when multiple domains are presented in the source (Leave-One-Out Domain Generalization, **LOODG**); and 2) when a single domain serves as the source (Single Source Domain Generalization, **SDG**). Unless specified, we report the mean and standard deviation of accuracies from three replications. Appendix D.1 provides information on the dataset and our implementations.

**Implementation of UDIM** To leverage a parameter $\theta$ exhibitng flat minima on $D_s$ during the minimization of $\mathcal{I}_s^\gamma(\theta)$, we perform a warm-up training on $\theta$ with the SAM loss on $D_s$. While keeping the total number of iterations consistent with other methods, we allocate initial iterations for warm-up based on the SAM loss. As a result, we expect to optimize Eq. 4 with a sufficiently wide region of $N_{s,\theta}^{\gamma,\rho}$ for sufficiently low $\gamma$.

The gradient variance, $\mathrm{Var}(\mathbf{G}_{D_s})$ in Eq. 10, necessitates the costly computation of per-sample gradients with respect to $\theta$. We utilize BackPACK (Dangel et al., 2020), which provides the faster computation of per-sample gradients. Also, we compute the gradient variance only for the classifier parameters, which is an efficient practice to improve the performance with low computational costs (Shin et al., 2023). Appendix D.1 specifies additional hyperparameter settings of UDIM.

**Baselines for Comparison** Since our approach, UDIM, is tested under both LOODG and SDG scenarios, we employed methods tailored for each scenario as baselines. These include strategies for robust optimization (Arjovsky et al., 2019; Sagawa et al., 2019) and augmentations for novel domain discovery (Zhang et al., 2018; Yun et al., 2019; Nam et al., 2021). Appendix D.2 enumer-

Table 3: Test accuracy for PACS. Each column represents test domain for **LOODG**, and train domain for **SDG**. * denotes that the performances are from its original paper. **Bold** indicates the best case of each column or improved performances when combined with the sharpness-based optimizers.

| Method | Leave-One-Out Source Domain Generalization | | | | | Single Source Domain Generalization | | | | |
|---|---|---|---|---|---|---|---|---|---|---|
| | Art | Cartoon | Photo | Sketch | Avg | Art | Cartoon | Photo | Sketch | Avg |
| Fishr* (Best among LOODG methods) | $88.4^*_{\pm0.2}$ | $78.7^*_{\pm0.7}$ | $97.0^*_{\pm0.1}$ | $77.8^*_{\pm2.0}$ | 85.5* | $75.9_{\pm1.7}$ | $81.1_{\pm0.7}$ | $46.9_{\pm0.7}$ | $57.2_{\pm4.4}$ | 65.3 |
| RIDG (Chen et al., 2023b) | $86.3_{\pm1.1}$ | $81.0_{\pm1.0}$ | $97.4_{\pm0.7}$ | $77.5_{\pm2.5}$ | 85.5 | $76.2_{\pm1.4}$ | $80.0_{\pm1.8}$ | $48.5_{\pm2.8}$ | $54.8_{\pm2.4}$ | 64.9 |
| ITTA (Chen et al., 2023a) | $87.9_{\pm1.4}$ | $78.6_{\pm2.7}$ | $96.2_{\pm0.2}$ | $80.7_{\pm2.2}$ | 85.8 | $78.4_{\pm1.5}$ | $79.8_{\pm1.3}$ | $56.5_{\pm3.7}$ | $60.7_{\pm0.9}$ | 68.8 |
| M-ADA | $85.5_{\pm0.7}$ | $80.7_{\pm1.5}$ | $97.2_{\pm0.5}$ | $78.4_{\pm1.4}$ | 85.4 | $78.0_{\pm1.1}$ | $79.5_{\pm1.2}$ | $47.1_{\pm0.4}$ | $55.7_{\pm0.5}$ | 65.1 |
| LTD | $85.7_{\pm1.9}$ | $79.9_{\pm0.9}$ | $96.9_{\pm0.5}$ | $83.3_{\pm0.5}$ | 86.4 | $76.8_{\pm0.7}$ | $82.5_{\pm0.4}$ | $56.2_{\pm2.5}$ | $53.6_{\pm1.4}$ | 67.3 |
| SAM | $86.8_{\pm0.6}$ | $79.6_{\pm1.4}$ | $96.8_{\pm0.1}$ | $80.2_{\pm0.7}$ | 85.9 | $77.7_{\pm1.1}$ | $80.5_{\pm0.6}$ | $46.7_{\pm1.1}$ | $54.2_{\pm1.5}$ | 64.8 |
| **UDIM w/ SAM** | $\mathbf{88.5}_{\pm0.1}$ | $\mathbf{86.1}_{\pm0.1}$ | $\mathbf{97.3}_{\pm0.1}$ | $\mathbf{82.7}_{\pm0.1}$ | **88.7** | $\mathbf{81.5}_{\pm0.1}$ | $\mathbf{85.3}_{\pm0.4}$ | $\mathbf{67.4}_{\pm0.6}$ | $\mathbf{64.6}_{\pm1.7}$ | **74.7** |
| SAGM | $85.3_{\pm2.5}$ | $80.9_{\pm1.1}$ | $97.1_{\pm0.4}$ | $77.8_{\pm0.5}$ | 85.3 | $78.9_{\pm1.2}$ | $79.8_{\pm1.0}$ | $44.7_{\pm1.8}$ | $55.6_{\pm1.1}$ | 64.8 |
| **UDIM w/ SAGM** | $\mathbf{88.9}_{\pm0.2}$ | $\mathbf{86.2}_{\pm0.3}$ | $\mathbf{97.4}_{\pm0.4}$ | $\mathbf{79.5}_{\pm0.8}$ | **88.0** | $\mathbf{81.6}_{\pm0.3}$ | $\mathbf{84.8}_{\pm1.2}$ | $\mathbf{68.1}_{\pm0.8}$ | $\mathbf{63.3}_{\pm0.9}$ | **74.5** |
| GAM | $85.5_{\pm0.6}$ | $81.1_{\pm1.0}$ | $96.4_{\pm0.2}$ | $81.0_{\pm1.7}$ | 86.0 | $79.1_{\pm1.3}$ | $79.7_{\pm0.9}$ | $46.3_{\pm0.6}$ | $56.6_{\pm1.1}$ | 65.4 |
| **UDIM w/ GAM** | $\mathbf{87.1}_{\pm0.9}$ | $\mathbf{86.3}_{\pm0.4}$ | $\mathbf{97.2}_{\pm0.1}$ | $\mathbf{81.8}_{\pm1.1}$ | **88.1** | $\mathbf{82.4}_{\pm0.9}$ | $\mathbf{84.2}_{\pm0.4}$ | $\mathbf{68.8}_{\pm0.8}$ | $\mathbf{64.0}_{\pm0.7}$ | **74.9** |

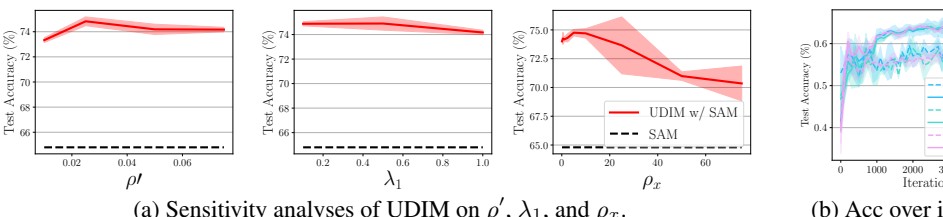

(a) Sensitivity analyses of UDIM on $\rho'$, $\lambda_1$, and $\rho_x$.      (b) Acc over iterations

Figure 3: (a) Sensitivity analyses of UDIM. (b) test accuracy plot of UDIM and sharpness-based approaches based on training iterations. Shaded regions represent standard deviation.

ates baselines for comparisons. For methods that leverage relationships between source domains, a straightforward application is not feasible in **SDG** scenarios. In such cases, we treated each batch as if it came from a different domain to measure experimental performance. ERM means the base model trained with standard classification loss. We also utilize the sharpness-based approaches as the baselines. Note that the first term of Eq. 4 (SAM loss on $D_s$) can be substituted with objectives for similar flatness outcomes, such as SAGM (Wang et al., 2023) and GAM (Zhang et al., 2023b).

## 4.2 CLASSIFICATION ACCURACIES ON VARIOUS BENCHMARK DATASETS

To assess the effectiveness of each method under unknown domains, we present the accuracy results on unknown target domains. Table 2 shows the results on CIFAR-10-C. The sharpness-based methods exhibit excellent performances compared to the other lines of methods. This underscores the importance of training that avoids overfitting to a specific domain. By adding UDIM (specifically, the inconsistency term of Eq. 4) to these SAM-based approaches, we consistently observed improved performances compared to the same approaches without UDIM.

Table 3 shows the results on the PACS dataset. Similar to the results in Table 2, UDIM consistently outperforms the existing baselines across each scenario. This improvement is particularly amplified in the single source scenario. Unlike the Leave-One-Out scenario, the single source scenario is more challenging as it requires generalizing to unknown domains using information from a single domain. These results emphasize the robust generalization capability of UDIM under unknown domains.

This section aims to examine the efficacy of UDIM by analyzing the sensitivity of each hyper-parameter used in UDIM's implementation. Additionally, We perform an ablation study by composing each part of the UDIM's objective in Eq. 12. Unless specified, each experiment is carried out in the Single source domain generalization scenario using the PACS dataset.

## 4.3 SENSITIVITY ANALYSES AND ABLATION STUDY

**Sensitivity Analyses** Figure 3 (a) shows the sensitivity of $\rho'$, $\lambda_1$ and $\rho_x$, which are main hyper-parameters of UDIM, under the feasible range of value lists. As a default setting of UDIM, we set $\rho'$=0.05, $\lambda_1$=1 and $\rho_x$=1. In implementation, we multiply $\rho_x$ by the unnormalized gradient of $x$. Therefore, the values presented in the ab-

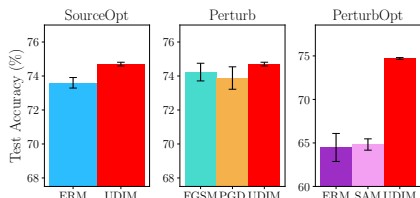

Figure 4: Ablation study of UDIM

lation study have a larger scale. Also, we compare the performances between SAM and UDIM w/ SAM to compare the effectiveness of UDIM objective. Each figure demonstrates that UDIM's performance remains robust and favorable, invariant to the changes in each hyper-parameter. Figure 3 (b) presents the test accuracies over training iterations while varying the sharpness-based approaches used alongside UDIM. Regardless of which method is used in conjunction, additional performance improvements over the iterations are observed compared to the original SAM variants.

**Ablation Studies** Figure 4 presents the ablation results of UDIM, which were carried out by replacing a subpart of the UDIM's objective with alternative candidates and subsequently assessing the performances. We conduct various ablations: 'SourceOpt' represents the optimization method for $D_s$, 'Perturb' indicates the perturbation method utilized to emulate unknown domains, and 'PerturbOpt' indicates the optimization for the perturbed dataset $\tilde{D}_s$. Appendix D.3 enumerates each ablation candidate and its implementation. UDIM, depicted by the red bar, consistently outperforms in all ablations, suggesting the effectiveness of our derived objective formulation.

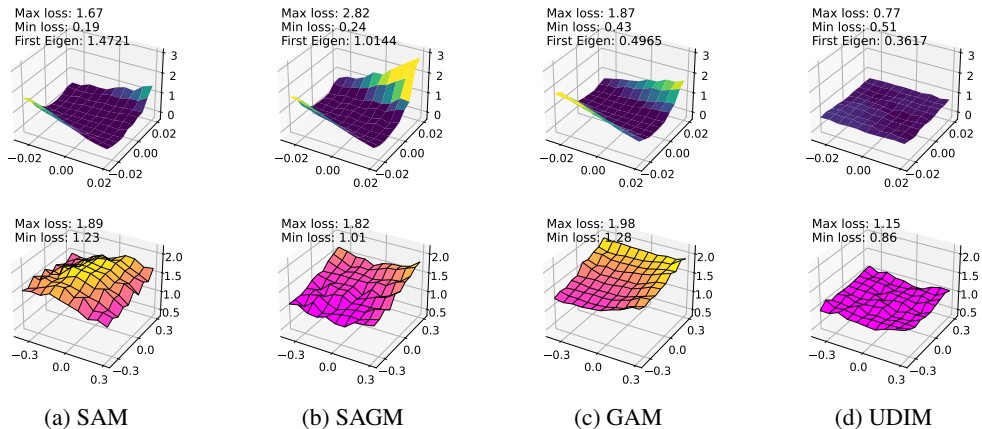

Figure 5: Sharpness plots for models trained using various methods: the upper plot shows sharpness on the perturbed parameter space, while the lower plot displays sharpness on the perturbed data space. The colormap of each row is normalized into the same scale for fair comparison.

## 4.4 SHARPNESS ANALYSES

Figure 1 claims that UDIM would reduce the suggested sharpness in both parameter space and data space. To support this claim with experiments, Figure 5 enumerates the sharpness plots for models trained with sharpness-based methods and those trained with UDIM. These figures are obtained by training models in the single source domain setting of the PACS dataset, and each plot is drawn utilizing target domain datasets, which are not utilized for training.

The top row of Figure 5 represents the measurement of sharpness in the parameter space by introducing a random perturbation to the parameter $\theta$. Additionally, to examine the sharpness in the perturbed data space of unobserved domains, the bottom side of Figure 5 illustrates sharpness based on the input-perturbed region of the target domain datasets. Current SAM variants struggle to maintain sufficient flatness within both the perturbed parameter space and data space. On the other hand, UDIM effectively preserves flatness in the perturbed parameter space and the data space of unknown domains. Within the region, the model trained using UDIM also exhibits a lower loss value compared to other methods. Through preserving flatness in each space, we confirm that the optimization with UDIM, both in parameter and data space, has practical effectiveness.

## 5 CONCLUSION

We introduce UDIM, a novel approach to minimize the discrepancy in the loss landscape between the source domain and unobserved domains. Combined with SAM variants, UDIM consistently improves generalization performance on unobserved domains. This performance is achieved by perturbing both domain and parameter spaces, where UDIM optimization is conducted based on the iterative update between the dataset and the parameter. Experimental results demonstrate accuracy gains, up to $9.9\%$ in some settings, by adopting UDIM in current sharpness-based approaches.

ACKNOWLEDGMENTS

This research was supported by AI Technology Development for Commonsense Extraction, Reasoning, and Inference from Heterogeneous Data(IITP) funded by the Ministry of Science and ICT(2022-0-00077).

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

# A    EXPLANATION OF SHARPNESS VARIANTS FOR DOMAIN GENERALIZATION

**Gradient norm-Aware Minimization (GAM)** Zhang et al. (2023b) introduces first-order flatness, which minimizes a maximal gradient norm within a perturbation radius, to regularize a stronger flatness than SAM. Accordingly, GAM seeks minima with uniformly flat curvature across all directions.

**Sharpness-Aware Gradient Matching (SAGM)** Wang et al. (2023) minimizes an original loss, the corresponding perturbed loss, and the gap between them. This optimization aims to identify a minima that is both flat and possesses a sufficiently low loss value. Interpreting the given formula, this optimization inherently regularizes the gradient alignment between the original loss and the perturbed loss.

**Flatness-Aware Minimization (FAM)** Zhang et al. (2023a) concurrently optimizes both zeroth-order and first-order flatness to identify flatter minima. To compute various sharpness metric on the different order, it incurs a higher computational cost.

# B    PROOFS AND DISCUSSIONS

## B.1    PROOF FOR THEOREM 3.2

First, we provide some theorem, definition, and assumptions needed to prove the Theorem 3.2.

**Theorem B.1.** *(Foret et al., 2020) For any $\rho > 0$ which satisfies $\mathcal{L}_{\mathscr{D}_e}(\theta) \leq \mathbb{E}_{\epsilon \sim p(\epsilon)} \mathcal{L}_{\mathscr{D}_e}(\theta + \epsilon)$, with probability at least $1 - \delta$ over realized dataset $D_e$ from $\mathscr{D}_e$ with $|D_e| = n$, the following holds under some technical conditions on $\mathcal{L}_{\mathscr{D}_e}(\theta)$:*

$$\mathcal{L}_{\mathscr{D}_e}(\theta) \leq \max_{\|\epsilon\|_2 \leq \rho} \mathcal{L}_{D_e}(\theta + \epsilon) + h_e\left(\frac{\|\theta\|_2^2}{\rho^2}\right),$$

*where $h_e : \mathbb{R}_+ \to \mathbb{R}_+$ is a strictly increasing function.*

**Definition B.2.** Let $\Theta_{\theta,\rho} = \{\theta' | \|\theta' - \theta\|_2 \leq \rho\}$ and $N_{e,\theta}^{\gamma,\rho} = \{\theta' \in \Theta_{\theta,\rho} | |\mathcal{L}_{D_e}(\theta') - \mathcal{L}_{D_e}(\theta)| \leq \gamma\}$.

**Assumption B.3.** $\mathcal{L}_{\mathscr{D}}(\theta) \leq \mathbb{E}_{\epsilon \sim p(\epsilon)} \mathcal{L}_{\mathscr{D}}(\theta + \epsilon)$ where $p(\epsilon) \sim \mathcal{N}(0, \sigma^2 I)$ for some $\sigma > 0$.

**Assumption B.4.** $\max_{e \in \mathcal{E}} \mathcal{L}_{D_e}(\theta') \geq \mathcal{L}_{D_s}(\theta')$ for all $\theta' \in N_{s,\theta}^{\gamma,\rho}$.

In practice, Assumption B.4 is acceptable. Contrary to the dataset $D_e$ from an unobserved domain $e$, $D_s$ is a source domain dataset provided to us and hence, available for training. Moreover, the region of $N_{s,\theta}^{\gamma,\rho}$ would be flat with respect to local minima, $\theta^*$, from the perspective of the source domain dataset. Therefore, the perturbed loss on the source domain dataset, $L_{D_s}(\theta')$, would be likely to have a sufficiently low value.

**Theorem B.5.** *For $\theta \in \Theta$ and arbitrary domain $e \in \mathcal{E}$, with probability at least $1 - \delta$ over realized dataset $D_e$ from $\mathscr{D}_e$ with $|D_e| = n$, the following holds under some technical conditions on $\mathcal{L}_{\mathscr{D}_e}(\theta)$.*

$$\mathcal{L}_{\mathscr{D}}(\theta) \leq \max_{\|\epsilon\|_2 \leq \rho} \mathcal{L}_{D_s}(\theta + \epsilon) + \left(1 - \frac{1}{|\mathcal{E}|}\right) \max_{e \in \mathcal{E}} \max_{\theta' \in N_{s,\theta}^{\gamma,\rho}} |\mathcal{L}_{D_e}(\theta') - \mathcal{L}_{D_s}(\theta')| + h\left(\frac{\|\theta\|_2^2}{\rho^2}\right) \quad (13)$$

*where $h : \mathbb{R}_+ \to \mathbb{R}_+$ is a strictly increasing function.*

*Proof.* For the derivation of Theorem 3.2, we assume the case of single source domain generalization, where $s$ represents a single domain in $\mathcal{E}$. It should be noted that the number of available source domain does not affect the validity of this proof because we can consider multiple source domains as a single source domain by $D_s = \cup_{i \in \mathcal{S}} D_i$. Based on the definition, $\mathcal{L}_{\mathscr{D}}(\theta) = \frac{1}{|\mathcal{E}|} \sum_{e \in \mathcal{E}} \mathcal{L}_{\mathscr{D}_e}(\theta) = \frac{1}{|\mathcal{E}|} \left( \mathcal{L}_{\mathscr{D}_s}(\theta) + \sum_{e \in \mathcal{E}, e \neq s} \mathcal{L}_{\mathscr{D}_e}(\theta) \right)$. From Theorem B.1, we can derive the generalization bound of a source domain $s$ as follows:

$$\mathcal{L}_{\mathscr{D}_s}(\theta) \leq \max_{\|\epsilon\|_2 \leq \rho} \mathcal{L}_{D_s}(\theta + \epsilon) + h_s\left(\frac{\|\theta\|_2^2}{\rho^2}\right) \quad (14)$$

where $h_s : \mathbb{R}_+ \rightarrow \mathbb{R}_+$ is a strictly increasing function. To define the upper bound of $\mathcal{L}_{\mathscr{D}}(\theta^*)$, we need to find the upper bound of the remaining term, $\sum_{e \in \mathcal{E}, e \neq s} \mathcal{L}_{\mathscr{D}_e}(\theta)$. Here, we introduce a parameter set $\Theta_{\rho'} := \arg\max_{\Theta_{\hat{\rho}} \subseteq N_{s,\theta}^{\gamma,\rho}} \hat{\rho}$, which is the largest $\rho'$-ball region around $\theta$ in $N_{s,\theta}^{\gamma,\rho}$. Then, we can construct inequality as follows:

$$\max_{\|\epsilon\|_2 \leq \rho'} \mathcal{L}_{D_e}(\theta + \epsilon) \leq \max_{\theta' \in N_{s,\theta}^{\gamma,\rho}} \mathcal{L}_{D_e}(\theta') \leq \max_{\|\epsilon\|_2 \leq \rho} \mathcal{L}_{D_e}(\theta + \epsilon) \tag{15}$$

This is because $\Theta_{\rho'} \subset N_{s,\theta}^{\gamma,\rho} \subset \Theta_{\theta,\rho}$. Similar to Foret et al. (2020); Kim et al. (2022), we make use of the following result from Laurent & Massart (2000):

$$z \sim \mathcal{N}(0, \sigma^2 I) \Rightarrow \|z\|_2^2 \leq k\sigma^2 \left(1 + \sqrt{\frac{\log n}{k}}\right)^2 \text{ with probability at least } 1 - \frac{1}{\sqrt{n}} \tag{16}$$

We set $\rho = \sigma(\sqrt{k} + \sqrt{\log n})$. Then, it enables us to connect expected perturbed loss and maximum loss as follows:

$$\mathbb{E}_{\epsilon \sim \mathcal{N}(0, \sigma^2 I)}\left[\mathcal{L}_{D_e}(\theta + \epsilon)\right] \leq (1 - \frac{1}{\sqrt{n}}) \max_{\|\epsilon\|_2 \leq \rho} \mathcal{L}_{D_e}(\theta + \epsilon) + \frac{1}{\sqrt{n}} l_{e,max} \tag{17}$$

Here, $l_{e,max}$ is the maximum loss bound when $\|z\|_2^2 \geq \rho^2$. Also, we introduce $\sigma'$ where $\rho' = \sigma'(\sqrt{k} + \sqrt{\log n})$. Then, similar to Eq. 17, we also derive the equation for $\rho'$ as follows:

$$\mathbb{E}_{\epsilon \sim \mathcal{N}(0, (\sigma')^2 I)}\left[\mathcal{L}_{D_e}(\theta + \epsilon)\right] \leq (1 - \frac{1}{\sqrt{n}}) \max_{\|\epsilon\|_2 \leq \rho'} \mathcal{L}_{D_e}(\theta + \epsilon) + \frac{1}{\sqrt{n}} l'_{e,max} \tag{18}$$

$$\leq (1 - \frac{1}{\sqrt{n}}) \max_{\theta' \in N_{s,\theta}^{\gamma,\rho}} \mathcal{L}_{D_e}(\theta') + \frac{1}{\sqrt{n}} l'_{e,max} \tag{19}$$

where $l'_{e,max}$ is the maximum loss bound when $\|z\|_2^2 \geq (\rho')^2$. Then, we have the below derivation by summing up all $e \neq s$ and using the fact that $l_{e,max} \leq l'_{e,max}$:

$$\frac{1}{(|\mathcal{E}| - 1)} \sum_{e \in \mathcal{E}, e \neq s} \mathbb{E}_{\epsilon \sim \mathcal{N}(0, \sigma'^2 I)}\left[\mathcal{L}_{D_e}(\theta + \epsilon)\right] \tag{20}$$

$$\leq (1 - \frac{1}{\sqrt{n}}) \frac{1}{(|\mathcal{E}| - 1)} \sum_{e \in \mathcal{E}, e \neq s} \max_{\theta' \in N_{s,\theta}^{\gamma,\rho}} \mathcal{L}_{D_e}(\theta') + \frac{1}{\sqrt{n}} \frac{1}{(|\mathcal{E}| - 1)} \sum_{e \in \mathcal{E}, e \neq s} l'_{e,max} \tag{21}$$

$$\leq (1 - \frac{1}{\sqrt{n}}) \max_{e \in \mathcal{E}} \max_{\theta' \in N_{s,\theta}^{\gamma,\rho}} \mathcal{L}_{D_e}(\theta') + \frac{1}{\sqrt{n}} \max_{e \in \mathcal{E}} l'_{e,max} \tag{22}$$

Using Assumption B.3 and PAC-Bayesian generalization bound (McAllester, 1999; Dziugaite & Roy, 2017; Foret et al., 2020), we find the upper bound of the sum of target domain losses.

$$\frac{1}{(|\mathcal{E}| - 1)} \sum_{e \in \mathcal{E}, e \neq s} \mathcal{L}_{\mathscr{D}_e}(\theta) \leq \frac{1}{(|\mathcal{E}| - 1)} \sum_{e \in \mathcal{E}, e \neq s} \mathbb{E}_{\epsilon \sim \mathcal{N}(0, (\sigma')^2 I)}[\mathcal{L}_{\mathscr{D}_e}(\theta + \epsilon)] \tag{23}$$

$$\leq \frac{1}{(|\mathcal{E}| - 1)} \sum_{e \in \mathcal{E}, e \neq s} \{\mathbb{E}_{\epsilon \sim \mathcal{N}(0, (\sigma')^2 I)}[\mathcal{L}_{D_e}(\theta + \epsilon)] + h_e(\frac{\|\theta\|_2^2}{\rho^2})\} \tag{24}$$

$$\leq (1 - \frac{1}{\sqrt{n}}) \max_{e \in \mathcal{E}} \max_{\theta' \in N_{s,\theta}^{\gamma,\rho}} \mathcal{L}_{D_e}(\theta') + \frac{1}{\sqrt{n}} \max_{e \in \mathcal{E}} l'_{e,max} + \frac{1}{(|\mathcal{E}| - 1)} \sum_{e \in \mathcal{E}, e \neq s} h_e(\frac{\|\theta\|_2^2}{\rho^2}) \tag{25}$$

$$\Rightarrow \sum_{e \in \mathcal{E}, e \neq s} \mathcal{L}_{\mathscr{D}_e}(\theta) \leq (|\mathcal{E}| - 1)(1 - \frac{1}{\sqrt{n}}) \max_{e \in \mathcal{E}} \max_{\theta' \in N_{s,\theta}^{\gamma,\rho}} \mathcal{L}_{D_e}(\theta') + \tilde{h}(\frac{\|\theta\|_2^2}{\rho^2}) \tag{26}$$

where $h_e, \tilde{h}$ are strictly increasing functions. We use the fact that the sum of strictly increasing functions is also a strictly increasing function. By integrating results of Eq. 14 and 26,

$$\frac{1}{|\mathcal{E}|}\Big(\mathcal{L}_{\mathscr{D}_s}(\theta) + \sum_{e \in \mathcal{E}, e \neq s} \mathcal{L}_{\mathscr{D}_e}(\theta)\Big) \leq \frac{1}{|\mathcal{E}|} \max_{\|\epsilon\|_2 \leq \rho} \mathcal{L}_{D_s}(\theta + \epsilon)$$
$$+ (1 - \frac{1}{|\mathcal{E}|})(1 - \frac{1}{\sqrt{n}}) \max_{e \in \mathcal{E}} \max_{\theta' \in N_{s,\theta}^{\gamma,\rho}} \mathcal{L}_{D_e}(\theta') + h(\frac{\|\theta\|_2^2}{\rho^2}) \tag{27}$$

where $h : \mathbb{R}_+ \to \mathbb{R}_+$ is a strictly increasing function. The first and second terms of RHS in the above inequality are upper bounded by the maximum perturbed loss for source domain and unknown domain inconsistency loss.

$$\frac{1}{|\mathcal{E}|} \max_{\|\epsilon\|_2 \leq \rho} \mathcal{L}_{D_s}(\theta + \epsilon) + (1 - \frac{1}{|\mathcal{E}|})(1 - \frac{1}{\sqrt{n}}) \max_{e \in \mathcal{E}} \max_{\theta' \in N_{s,\theta}^{\gamma,\rho}} \mathcal{L}_{D_e}(\theta') \tag{28}$$

$$\leq \frac{1}{|\mathcal{E}|} \max_{\|\epsilon\|_2 \leq \rho} \mathcal{L}_{D_s}(\theta + \epsilon) + (1 - \frac{1}{|\mathcal{E}|}) \max_{e \in \mathcal{E}} \max_{\theta' \in N_{s,\theta}^{\gamma,\rho}} \mathcal{L}_{D_e}(\theta') \tag{29}$$

$$= \max_{\|\epsilon\|_2 \leq \rho} \mathcal{L}_{D_s}(\theta + \epsilon) + (1 - \frac{1}{|\mathcal{E}|}) \Big( \max_{e \in \mathcal{E}} \max_{\theta' \in N_{s,\theta}^{\gamma,\rho}} \mathcal{L}_{D_e}(\theta') - \max_{\|\epsilon\|_2 \leq \rho} \mathcal{L}_{D_s}(\theta + \epsilon) \Big) \tag{30}$$

$$\leq \max_{\|\epsilon\|_2 \leq \rho} \mathcal{L}_{D_s}(\theta + \epsilon) + (1 - \frac{1}{|\mathcal{E}|}) \Big( \max_{e \in \mathcal{E}} \max_{\theta' \in N_{s,\theta}^{\gamma,\rho}} \mathcal{L}_{D_e}(\theta') - \max_{\theta' \in N_{s,\theta}^{\gamma,\rho}} \mathcal{L}_{D_s}(\theta') \Big) \tag{31}$$

$$\leq \max_{\|\epsilon\|_2 \leq \rho} \mathcal{L}_{D_s}(\theta + \epsilon) + (1 - \frac{1}{|\mathcal{E}|}) \max_{e \in \mathcal{E}} \max_{\theta' \in N_{s,\theta}^{\gamma,\rho}} (\mathcal{L}_{D_e}(\theta') - \mathcal{L}_{D_s}(\theta')) \tag{32}$$

$$= \max_{\|\epsilon\|_2 \leq \rho} \mathcal{L}_{D_s}(\theta + \epsilon) + (1 - \frac{1}{|\mathcal{E}|}) \max_{e \in \mathcal{E}} \max_{\theta' \in N_{s,\theta}^{\gamma,\rho}} |\mathcal{L}_{D_e}(\theta') - \mathcal{L}_{D_s}(\theta')| \tag{33}$$

The last equality comes from the Assumption B.4. To sum up, we can derive the upper bound of the population loss for whole domain using the maximum perturbed loss for source domain and unknown domain inconsistency loss with weight decay term.

$$\mathcal{L}_{\mathscr{D}}(\theta) \leq \max_{\|\epsilon\|_2 \leq \rho} \mathcal{L}_{D_s}(\theta + \epsilon) + (1 - \frac{1}{|\mathcal{E}|}) \max_{e \in \mathcal{E}} \max_{\theta' \in N_{s,\theta}^{\gamma,\rho}} |\mathcal{L}_{D_e}(\theta') - \mathcal{L}_{D_s}(\theta')| + h(\frac{\|\theta\|_2^2}{\rho^2}) \tag{34}$$

$\square$

## B.2 Detailed Explanation on Approximation

Here, we show the full derivation of Eq. 8 and 9 as follows.

$$\max_{\theta' \in N_{s,\theta}^{\gamma,\rho}} \Big( \mathcal{L}_{\tilde{D}_s}(\theta') - \mathcal{L}_{D_s}(\theta') \Big) \approx \max_{\theta' \in N_{s,\theta}^{\gamma,\rho}} \mathcal{L}_{\tilde{D}_s}(\theta') - \mathcal{L}_{D_s}(\theta) + \gamma' \tag{35}$$

$$\underset{\text{2nd Taylor}}{\approx} \mathcal{L}_{\tilde{D}_s}(\theta) - \mathcal{L}_{D_s}(\theta) + \rho' \|\nabla_\theta \mathcal{L}_{\tilde{D}_s}(\theta)\|_2 + \max_{\theta' \in N_{s,\theta}^{\gamma,\rho}} \frac{1}{2}\theta'^\top \mathbf{H}_{\tilde{D}_s} \theta' \tag{36}$$

$$= \Big( \mathcal{L}_{\tilde{D}_s}(\theta) - \mathcal{L}_{D_s}(\theta) \Big) + \rho' \|\nabla_\theta \mathcal{L}_{\tilde{D}_s}(\theta)\|_2 + \gamma \max_i \lambda_i^{\tilde{D}_s} / \lambda_i^{D_s} \tag{37}$$

## B.3 Discussion on Hessian matrix and Gradient Variance

**Hessian Matching**   This section first discusses how the Hessian matrix matching between two different datasets could be substituted by the gradient variance matching for the respective datasets. It should be noted that we follow the motivation and derivation of Rame et al. (2022), and this section just re-formalize the derivations based on our notations. $\mathbf{g}_i$ is a per-sample gradient for $i$-th sample; and $\mathbf{G}_D = \{\mathbf{g}_i\}_{i=1}^{|D|}$ is a set of per-sample gradient for $x_i \in D$. Accordingly, the variance of $\mathbf{G}_D$, which we denote as $\text{Var}(\mathbf{G}_D)$, is calculated as $\text{Var}(\mathbf{G}_D) = \frac{1}{|D|-1} \sum_{i=1}^{|D|} (\mathbf{g}_i - \bar{\mathbf{g}})^2$. We first revisit the Eq. 37, which is our intermediate objective as follows:

$$\Big( \mathcal{L}_{\tilde{D}_s}(\theta) - \mathcal{L}_{D_s}(\theta) \Big) + \rho' \|\nabla_\theta \mathcal{L}_{\tilde{D}_s}(\theta)\|_2 + \gamma \max_i \lambda_i^{\tilde{D}_s} / \lambda_i^{D_s} \tag{38}$$

The last term of Eq. 38, $\max_i \lambda_i^{\tilde{D}_s} / \lambda_i^{D_s}$, refers to the maximum eigenvalue ratio of the Hessian matrices between two different datasets, $\tilde{D}_s$ and $D_s$. This ratio is minimized when Hessian matrices of $\tilde{D}_s$ and $D_s$ becomes equivalent. Then, $\max_i \lambda_i^{\tilde{D}_s} / \lambda_i^{D_s}$ is approximated to the hessian matrix matching between $\tilde{D}_s$ and $D_s$ as $\|\mathbf{H}_{\tilde{D}_s} - \mathbf{H}_{D_s}\|_2$. Computing the full Hessian matrix in overparameterized networks is computationally challenging. Therefore, we express the formula using the diagonalized Hessian matrix, denoted as $\hat{\mathbf{H}}_{D_s}$, which results in $\|\hat{\mathbf{H}}_{\tilde{D}_s} - \hat{\mathbf{H}}_{D_s}\|_2$.

Let the Fisher Information Matrix (Rame et al., 2022) as $\boldsymbol{F} = \sum_{i=1}^{n} \mathbb{E}_{\hat{y} \sim P_\theta(\cdot|x_i)} \left[ \nabla_\theta \log p_\theta(\hat{y}|x_i) \nabla_\theta \log p_\theta(\hat{y}|x_i)^\top \right]$, where $p_\theta(\cdot|x_i)$ is the density of $f_\theta$ on a input instance $x$. Fisher Information Matrix (FIM) approximates the Hessian $\mathbf{H}$ with theoretically probably bounded errors under mild assumptions Schraudolph (2002). Then, diagonalized Hessian matrix matching between $\tilde{D}_s$ and $D_s$, $\|\hat{\mathbf{H}}_{\tilde{D}_s} - \hat{\mathbf{H}}_{D_s}\|_2$ could be replaced by $\|\hat{\boldsymbol{F}}_{\tilde{D}_s} - \hat{\boldsymbol{F}}_{D_s}\|_2$, where $\hat{\boldsymbol{F}}$ also denotes the diagonalized version of $\boldsymbol{F}$. Empirically, $\hat{\boldsymbol{F}}$ is equivalent to the gradient variance of the trained model, $f_\theta$. This finally confirms the validation of our objective, gradient variance difference between $\tilde{D}_s$ and $D_s$ as $\|\text{Var}(\mathbf{G}_{\tilde{D}_s}) - \text{Var}(\mathbf{G}_{D_s})\|_2$. Table 2 on the main paper of Rame et al. (2022) empirically supports that the similarity between Hessian diagonals and gradient variances is over 99.99%.

**Loss Matching**    Matching the gradient variances for all parameters of our model, $f_\theta$, incurs significant computational overhead. In this study, we restrict the gradient variance matching to a subset of entire parameters, specifically selecting the classifier parameters. In this section, we demonstrate that by matching gradient variances of two different datasets based on the classifier parameters, it inherently achieve the loss matching across those datasets.

For simplicity in notation, we refer an arbitrary domain index as $e$. Let $x_e^i$ represent the $i$-th sample and $y_e^i$ be its corresponding target class label. We denote $z_e^i \in \mathbb{R}^d$ as the features for this $i$-th sample from domain $e$. The associated classifier layer $W$ is characterized by weights $\{w_k\}_{k=1}^d$ and bias $b$. First, we assume the mean squared error as our loss function for our analysis. For the $i$-th sample, the gradient of the loss with respect to $b$ is given by $\nabla_b \ell(f_\theta(x_e^i), y_e^i) = (f_\theta(x_e^i) - y_e^i)$. Hence, the gradient variance based on the parameter $b$ for domain $e$ is given by: $\text{Var}(\mathbf{G}_{D_e}^b) = \frac{1}{n_e} \sum_{i=1}^{n_e} (f_\theta(x_e^i) - y_e^i)^2$, which directly aligns with the mean squared error (MSE) between the predictions and the target labels in domain $e$. Considering our objective, $\|\text{Var}(\mathbf{G}_{\tilde{D}_s}) - \text{Var}(\mathbf{G}_{D_s})\|_2$, the gradient variance matching on $b$ could be recognized as mean squared error loss matching, which is $\|\frac{1}{|\tilde{D}_s|} \sum_{(x^i,y^i) \in \tilde{D}_s} (f_\theta(x^i) - y^i)^2 - \frac{1}{|D_s|} \sum_{(x^j,y^j) \in D_s} (f_\theta(x^j) - y^j)^2\|_2$.

**Analysis on the remaining term**    We also investigate the gradient variance matching based on $\{w_k\}_{k=1}^d \in W$, which are remaining part of the classifier parameter $W$. The gradients with respect to the $w_k$ is derived as $\nabla_{w_k} \ell(y_e^i, \hat{y}_e^i) = (\hat{y}_e^i - y_e^i) z_e^{i,k}$. Thus, the uncentered gradient variance in $w_k$ for domain $e$ is: $\text{Var}(\mathbf{G}_{D_e}^{w_k}) = \frac{1}{n_e} \sum_{i=1}^{n_e} \left( (\hat{y}_e^i - y_e^i) z_e^{i,k} \right)^2$. Different from the case of $b$, $\text{Var}(\mathbf{G}_{D_e}^{w_k})$ adds a weight $z_e^{i,k}$ on the mean squared error. As $z_e^{i,k}$ act as a weight, gradient variance matching on $w_k$ still learns toward matching the MSE loss between two different datasets.

## C  ALGORITHM OF UDIM

Here, we present the algorithm of UDIM as follows.

---

**Algorithm 1:** Training algorithm of UDIM w/ SAM

---

**Input:** Source dataset $D_s$; perturbation threshold for model parameter $\theta$ and data, $\rho'$ and $\rho_x$;
      learning rate $\eta$; warmup ratio for source domain flatness, $p$; number of total training
      iterations, $N$; Other hyperpameters.

**Output:** Trained model $f_\theta(x)$

**for** $t = 1, ..., N/p$ **do**
   |   Warmup $f_\theta(x)$ with SAM optimization as Eq. 1
**end**

**for** $t = N/p, ..., N$ **do**

    Define $D_B = \{(x_i, y_i)\}_{i=1}^{|B|}$ i.i.d. sampled from $D_s$

    Make $\tilde{D}_B = \{(\tilde{x}_i, y_i)\}_{i=1}^{|B|}$, with $\tilde{x}_i$ for all $i$ as $\tilde{x}_i = x_i + \rho_x \dfrac{\nabla_{x_i}\left(\ell(x,\theta_t) + \rho'\|\nabla_{\theta_t}\ell(x,\theta_t)\|_2\right)}{\left\|\nabla_{x_i}\left(\ell(x,\theta_t)+\rho'\|\nabla_{\theta_t}\ell(x,\theta_t)\|_2\right)\right\|_2}$

    Update $f_\theta(x)$ by $\theta_{t+1} \leftarrow \theta_t - \eta\nabla_{\theta_t}\Big( \max_{\|\epsilon\|_2 \le \rho} \mathcal{L}_{D_B}(\theta_t + \epsilon) + \rho'\|\nabla_{\theta_t}\mathcal{L}_{\tilde{D}_B}(\theta_t)\|_2 +$

    $\|\text{Var}(\mathbf{G}_{\tilde{D}_B}) - \text{Var}(\mathbf{G}_{D_B})\|_2 + \lambda_2\|\theta_t\|_2\Big)$

**end**

---

We represent the algorithm of UDIM with SAM as a default setting. It should be noted that our method can be orthogonally utilized with other sharpness-based optimization methods.

## D  EXPERIMENT

### D.1  IMPLEMENTATION DETAILS

**Dataset Explanation**

- PACS (Li et al., 2017) comprises of four domains, which are photos, arts, cartoons and sketches. This dataset contains 9,991 images. It consists of 7 classes.
- OfficeHome (Venkateswara et al., 2017) includes four domains, which are art, clipart, product and real. This dataset contains 15,588 images. It consists of 65 classes.
- DomainNet (Peng et al., 2019) consists of six domains, which are clipart, infograph, painting, quickdraw, real and sketch. This dataset contains 586,575 images. It consists of 345 classes.
- CIFAR-10-C (Hendrycks & Dietterich, 2019) has been utilized to evaluate the robustness of a trained classifier. Images are corrupted from CIFAR10 (Krizhevsky et al., 2009) test dataset under 5 levels of severity. Corruption types include, brightness, contrast, defocus-blur, elastic-transform, fog, frost, gaussian-blur, gaussian-noise, glass-blur, impulse-noise, jpeg-compression, motion-blur, pixelate, saturate, shot-noise, snow, spatter, speckle-noise, zoom-blur total of 19 types.

**Network Architecture and Optimization**    We use ResNet-18 (He et al., 2016) for CIFAR-10-C and ResNet-50 (He et al., 2016) for other datasets pretrained on ImageNet (Deng et al., 2009) and use Adam (Kingma & Ba, 2014) optimizer basically. learning rate is set as $3 \times 10^{-5}$ following Wang et al. (2023). For calculating the gradient related materials, e.g. Hessian, we utilize BackPACK (Dangel et al., 2020) package.

**Experimental settings**    For PACS and OfficeHome, we trained for total of 5,000 iterations. For DomainNet, we trained for 15,000 iterations. For CIFAR-10, since it usually trains for 100 epochs, we translate it to iterations, which becomes total of $781 \times 100 = 78,100$ iterations. Unless specified,

we use batch size as 32 for PACS, OfficeHome, and DomainNet and 64 for CIFAR-10-C. For other hyperparameters, we follow the experiment settings of Wang et al. (2023) unless specified. Although our method mainly focuses on the domain generalization, our concept could be also effectively utilized for domain adaptation Csurka (2017) and open-set domain adaptation Jang et al. (2022).

**Hyperparameter setting of UDIM**   The main hyperparameters of UDIM is $\rho, \rho^{'}$, $\lambda_1$ and $\rho_x$. Throughout all experiments, we set $\rho$=0.05 without any hyperparameter tuning. For $\rho^{'}$, we used values [0.01, 0.025, 0.05], and in most experiments, the value 0.05 consistently showed good performance. It should be noted that a warm-up using the SAM loss is required before the full UDIM optimization to ensure that $\rho^{'}$=0.05 can be utilized validly. For both $\lambda_1$ and $\rho_x$, we used values in the range [1,10] and reported the best performances observed among these results. Our methodology applies perturbations to each instance of the original source domain dataset, effectively doubling the number of unique instances in a single batch compared to experiments for the baselinse. As a result, we utilized half the batch size of other baselines.

**Evaluation Detail**   For reporting the model performance, model selection criterion is important. We get the test performance whose accuracy for source validation dataset is best. For PACS and OfficeHome, we evaluated every 100 iterations and for DomainNet, we evaluated every 1,000 iterations for Leave-One-Out Source Domain Generalization and every 5,000 iterations for Single Source Domain Generalization.

## D.2   BASELINE DESCRIPTION

In this paragraph, we explain baselines that we used for comparison. Specifically, we compare our method with (1) methods whose objectives are mainly related to Leave-One Out Source Domain Generalization, (2) methods which are mainly modeled for Single Source Domain Generalization, and (3) sharpness-aware minimization related methods, as we reported in tables repeatedly.

**IRM** (Arjovsky et al., 2019) tries to learn a data representation such that the optimal classifier matches for all training distributions. Specifically, it minimizes the empirical risk and the regularization term, the multiplication of samples' gradients, to motivate the invariance of the predictor.

**GroupDRO** (Sagawa et al., 2019) minimizes the loss by giving different weight to each domain. Weight term for each domain is proportional to the domain's current loss.

**OrgMixup** (Zhang et al., 2018) represents the naive mixup technique which is generally utilized in machine learning community to boost generalization.

**Mixup** (Yan et al., 2020) is a mixup among domains.

**Cutmix** (Yun et al., 2019) is another skill which is widely used in machine learning community to boost generalization. Specifically, it mixes up parts of inputs randomly by pixel-wise.

**Mixstyle** (Zhou et al., 2021) mix up the statistics (specifically, mean and standard deviation) of the feature. The mixed feature statistics are applied to the style-normalized input. We did not consider the domain label.

**MTL** (Blanchard et al., 2021) considers the exponential moving average (EMA) of features.

**MLDG** (Li et al., 2018a) is a meta learning based method for domain generalization. Specifically, it simulates the domain shift between train and test during training procedure by synthesizing virtual testing domains within each mini-batch. Then it optimizes meta loss using the synthesized dataset.

**MMD** (Li et al., 2018b) minimizes the discrepancy of feature distributions in a every domain pairwise manner, while minimizing the empirical risk for source domains.

**CORAL** (Sun & Saenko, 2016) is similar to **MMD**. However, while **MMD** employs the gaussian kernel to measure the feature discrepancy, **CORAL** aligns the second-order statistics between different distributions with a nonlinear transformation. This alignment is achieved by matching the correlations of layer activations in deep neural networks.

**SagNet** (Nam et al., 2021) disentangles style features from class categories to prevent bias. Specifically, it makes two networks, content network and style network, and trains both networks to be

invariant to other counterpart by giving randomized features (updating the content network with randomized styled features and vice versa).

**ARM** (Zhang et al., 2021) represents adaptive risk minimization. Specifically, it makes an adaptive risk representing context.

**DANN** represents Domain Adversarial Neural Networks, and it iteratively trains a discriminator which discriminates domain and a featurizer to learn a feature which becomes invariant to domain information.

**CDANN** is class conditional version of DANN.

**VREx** (Krueger et al., 2021) controls the discrepancy between domains by minimizing the variance of loss between domains.

**RSC** (Huang et al., 2020) challenges the dominant features of training domain (by masking some specific percentage of dominant gradient), so it can focus on label-related domain invariant features.

**Fishr** (Rame et al., 2022) approximates the hessian as the variance of gradient matrix, and they align the gradient variance of each domain.

**M-ADA** (Qiao et al., 2020a) perturbs input data to simulate the unseen domain data, yet with adequate regularization not to make the data be too far from the original one. The adversarial perturbation direction is affected by the wasserstein autoencoder. Note that this method is specifically designed for Single source domain generalization.

**LTD** (Wang et al., 2021a) perturbs source domain data with augmentation network, maximize the mutual information between the original feature and the perturbed feature so that the perturbed feature is not too far from the original feature (with contrastive loss), and maximize likelihood of the original feature. Note that this method is also specifically designed for Single source domain generalization.

**SAM** (Foret et al., 2020) is an optimization technique to consider the sharpness of loss surface. It first perturbs parameter to its worst direction, gets gradient and update the calculated gradient at the original parameter point.

**SAGM** (Wang et al., 2023) minimizes an original loss, the corresponding perturbed loss, and the gap between them. This optimization aims to identify a minima that is both flat and possesses a sufficiently low loss value. Interpreting the given formula, this optimization inherently regularizes the gradient alignment between the original loss and the perturbed loss.

**GAM** (Zhang et al., 2023b) introduces first-order flatness, which minimizes a maximal gradient norm within a perturbation radius, to regularize a stronger flatness than SAM. Accordingly, GAM seeks minima with uniformly flat curvature across all directions.

**RIDG** (Chen et al., 2023b) presents a new approach in deep neural networks focusing on decision-making in the classifier layer, diverging from the traditional emphasis on features. It introduces a 'rationale matrix', derived from the relationship between features and weights, to guide decisions for each input. A novel regularization term is proposed to align each sample's rationale with the class's mean, enhancing stability across samples and domains.

**ITTA** (Chen et al., 2023a) proposes an Improved Test-Time Adaptation (ITTA) method for domain generalization. ITTA uses a learnable consistency loss for the TTT task to better align with the main prediction task and introduces adaptive parameters in the model, recommending updates solely during the test phase. This approach aims to address the issues of auxiliary task selection and parameter updating in test-time training.

## D.3 ABLATION

Figure 4 of the main paper presents the ablation results of UDIM, which were carried out by replacing a subpart of the UDIM's objective with alternative candidates and subsequently assessing the performances. This section enumerates enumerates each ablation candidate and its implementation. We conduct various ablations: 'SourceOpt' represents the optimization method for $D_s$, 'Perturb' indicates the perturbation method utilized to emulate unknown domains, and 'PerturbOpt' indicates

the optimization for the perturbed dataset $\tilde{D}_s$. It should be noted that each ablation means that only the specific part is substituted, while keeping the other parts of UDIM unchanged.

**SourceOpt** The optimization for the source domain dataset in UDIM is originally based on the SAM loss, which is $\max_{\|\epsilon\|_2 \leq \rho} \mathcal{L}_{D_s}(\theta + \epsilon)$. To verify the significance of flatness modeling, we replaced the original optimization with simple ERM, which we refer to as the **ERM** ablation.

**Perturb** The perturbation process of UDIM is conducted based on Eq. 11. We substituted the perturbation method in Eq. 11 with traditional adversarial attack techniques, which are the cases of **FGSM** (Goodfellow et al., 2014) and **PGD** (Madry et al., 2017), to compare their performance outcomes.

**PerturbOpt** Lastly, to observe the ablation for inconsistency minimization between the perturbed domain dataset $\tilde{D}_s$ and $D_s$, we replaced the optimization in Eq. 10 with both ERM and SAM-based optimizations. Each case in the PerturbOpt ablation is denoted as **ERM** or **SAM**.

## D.4 ADDITIONAL RESULTS

In this section, we report more results that we did not report in the main paper due to the space issue.

Table 4 shows the model performance of total baselines (At the table 3 of the main paper, there are only the model performance of some baselines). As we can see in the table, our method, UDIM, consistently improves SAM-based optimization variants and show best performances for each column. We mark - for training failure case (when the model performance is near 1%).

Table 4: Test accuracy for PACS. For **Leave-One-Out Source Domain Generalization**, each column represents test domain, and train domain for **Single Source Domain Generalization**. * denotes performances are from its original paper considering LOODG. For SDG scenario, we generated the experiment results for all baselines. **Bold** indicates the best case of each column or improved performances when combined with the respective sharpness-based optimizers.

| Method | Leave-One-Out Source Domain Generalization | | | | | Single Source Domain Generalization | | | | |
|---|---|---|---|---|---|---|---|---|---|---|
| | Art | Cartoon | Photo | Sketch | Avg | Art | Cartoon | Photo | Sketch | Avg |
| ERM | $86.9_{\pm2.3}$ | $79.5_{\pm1.5}$ | $96.6_{\pm0.5}$ | $78.2_{\pm4.1}$ | 85.3 | $79.9_{\pm0.9}$ | $79.9_{\pm0.8}$ | $48.1_{\pm5.8}$ | $59.6_{\pm1.1}$ | 66.9 |
| IRM* | $85.0^*_{\pm1.6}$ | $77.6^*_{\pm0.9}$ | $96.7^*_{\pm0.3}$ | $78.5^*_{\pm2.6}$ | 84.4* | $73.3_{\pm1.5}$ | $77.8_{\pm2.3}$ | $46.9_{\pm0.8}$ | $49.7_{\pm3.0}$ | 61.9 |
| GroupDRO | $84.8_{\pm2.2}$ | $79.4_{\pm1.2}$ | $97.3_{\pm0.3}$ | $75.8_{\pm1.0}$ | 84.3 | $79.0_{\pm0.5}$ | $79.0_{\pm0.6}$ | $42.0_{\pm2.9}$ | $60.8_{\pm3.9}$ | 65.2 |
| OrgMixup | $87.7_{\pm0.3}$ | $77.4_{\pm1.3}$ | $97.6_{\pm0.3}$ | $76.3_{\pm0.9}$ | 84.8 | $74.5_{\pm1.1}$ | $79.8_{\pm0.4}$ | $46.8_{\pm2.1}$ | $55.5_{\pm1.9}$ | 64.1 |
| Mixup | $86.9_{\pm1.3}$ | $78.2_{\pm0.7}$ | $97.8_{\pm0.4}$ | $73.7_{\pm2.9}$ | 84.2 | $77.4_{\pm1.4}$ | $80.0_{\pm1.2}$ | $47.3_{\pm1.6}$ | $58.2_{\pm1.2}$ | 65.7 |
| CutMix | $80.5_{\pm0.7}$ | $75.7_{\pm1.4}$ | $97.0_{\pm0.5}$ | $74.8_{\pm1.7}$ | 82.0 | $71.1_{\pm0.5}$ | $76.4_{\pm3.1}$ | $37.7_{\pm0.3}$ | $50.4_{\pm4.0}$ | 58.9 |
| Mixstyle | $84.4_{\pm2.3}$ | $80.4_{\pm0.6}$ | $95.6_{\pm0.1}$ | $80.5_{\pm1.1}$ | 85.2 | $78.1_{\pm2.8}$ | $78.8_{\pm1.1}$ | $56.1_{\pm3.9}$ | $54.7_{\pm2.9}$ | 66.9 |
| MTL | $85.4_{\pm2.2}$ | $78.8_{\pm2.2}$ | $96.5_{\pm0.2}$ | $74.4_{\pm2.0}$ | 83.8 | $76.7_{\pm1.2}$ | $78.7_{\pm1.7}$ | $44.7_{\pm2.0}$ | $59.5_{\pm1.4}$ | 64.9 |
| MLDG | $87.7_{\pm0.6}$ | $77.5_{\pm0.7}$ | $96.6_{\pm0.6}$ | $75.3_{\pm1.9}$ | 84.3 | - | - | - | - | - |
| MMD* | $84.5^*_{\pm0.6}$ | $79.7^*_{\pm0.7}$ | $97.5^*_{\pm0.4}$ | $78.1^*_{\pm1.3}$ | 85.0* | $75.4_{\pm1.1}$ | $80.1_{\pm0.5}$ | $45.2_{\pm1.2}$ | $58.2_{\pm0.6}$ | 64.7 |
| CORAL* | $87.7^*_{\pm0.6}$ | $79.2^*_{\pm1.1}$ | $97.6^*_{\pm0.0}$ | $79.4^*_{\pm0.7}$ | 86.0* | $76.3_{\pm0.8}$ | $79.2_{\pm2.0}$ | $45.9_{\pm1.7}$ | $57.0_{\pm1.4}$ | 64.6 |
| SagNet | $87.1_{\pm1.1}$ | $78.0_{\pm1.9}$ | $96.8_{\pm0.2}$ | $78.4_{\pm1.4}$ | 85.1 | $77.4_{\pm0.0}$ | $78.9_{\pm1.8}$ | $47.6_{\pm2.4}$ | $56.4_{\pm4.0}$ | 65.1 |
| ARM | $86.4_{\pm0.1}$ | $78.8_{\pm0.6}$ | $96.1_{\pm0.1}$ | $75.1_{\pm3.3}$ | 84.1 | $76.2_{\pm0.5}$ | $75.5_{\pm4.0}$ | $45.2_{\pm5.7}$ | $61.9_{\pm2.0}$ | 64.7 |
| DANN* | $85.9^*_{\pm0.5}$ | $79.9^*_{\pm1.4}$ | $97.6^*_{\pm0.2}$ | $75.2^*_{\pm2.8}$ | 84.6* | $79.0_{\pm1.4}$ | $76.5_{\pm2.0}$ | $48.7_{\pm2.1}$ | $57.9_{\pm4.7}$ | 65.5 |
| CDANN* | $84.0^*_{\pm0.9}$ | $78.5^*_{\pm1.5}$ | $97.0^*_{\pm0.4}$ | $71.8^*_{\pm3.9}$ | 82.8* | $78.5_{\pm1.5}$ | $78.7_{\pm2.0}$ | $48.3_{\pm3.1}$ | $56.9_{\pm2.2}$ | 65.6 |
| VREx | $87.2_{\pm0.5}$ | $77.8_{\pm0.8}$ | $96.8_{\pm0.3}$ | $75.2_{\pm3.4}$ | 84.3 | $75.3_{\pm2.1}$ | $80.2_{\pm0.4}$ | $44.9_{\pm2.8}$ | $56.8_{\pm2.6}$ | 64.3 |
| RSC | $81.0_{\pm0.7}$ | $77.6_{\pm1.0}$ | $95.3_{\pm0.8}$ | $75.0_{\pm1.4}$ | 82.2 | $68.9_{\pm2.3}$ | $70.6_{\pm3.6}$ | $41.1_{\pm3.1}$ | $45.9_{\pm3.1}$ | 56.6 |
| Fishr* | $88.4^*_{\pm0.2}$ | $78.7^*_{\pm0.7}$ | $97.0^*_{\pm0.1}$ | $77.8^*_{\pm2.0}$ | 85.5* | $75.9_{\pm1.7}$ | $81.1_{\pm0.7}$ | $46.9_{\pm0.7}$ | $57.2_{\pm4.4}$ | 65.3 |
| RIDG | $86.3_{\pm1.1}$ | $81.0_{\pm1.0}$ | $97.4_{\pm0.7}$ | $77.5_{\pm2.5}$ | 85.5 | $76.2_{\pm1.4}$ | $80.0_{\pm1.8}$ | $48.5_{\pm2.8}$ | $54.8_{\pm2.4}$ | 64.9 |
| ITTA | $87.9_{\pm1.4}$ | $78.6_{\pm2.7}$ | $96.2_{\pm0.2}$ | $80.7_{\pm2.2}$ | 85.8 | $78.4_{\pm1.5}$ | $79.8_{\pm1.3}$ | $56.5_{\pm3.7}$ | $60.7_{\pm0.9}$ | 68.8 |
| M-ADA | $85.5_{\pm0.7}$ | $80.7_{\pm1.5}$ | $97.2_{\pm0.5}$ | $78.4_{\pm1.4}$ | 85.4 | $78.0_{\pm1.1}$ | $79.5_{\pm1.2}$ | $47.1_{\pm0.4}$ | $55.7_{\pm0.5}$ | 65.1 |
| LTD | $85.7_{\pm1.9}$ | $79.9_{\pm0.9}$ | $96.9_{\pm0.5}$ | $83.3_{\pm0.5}$ | 86.4 | $76.8_{\pm0.7}$ | $82.5_{\pm0.4}$ | $56.2_{\pm2.5}$ | $53.6_{\pm1.4}$ | 67.3 |
| SAM | $86.8_{\pm0.6}$ | $79.6_{\pm1.4}$ | $96.8_{\pm0.1}$ | $80.2_{\pm0.7}$ | 85.9 | $77.7_{\pm1.1}$ | $80.5_{\pm0.6}$ | $46.7_{\pm1.1}$ | $54.2_{\pm1.5}$ | 64.8 |
| **UDIM w/ SAM** | **$88.5_{\pm0.1}$** | **$86.1_{\pm0.1}$** | **$97.3_{\pm0.1}$** | **$82.7_{\pm0.1}$** | **88.7** | **$81.5_{\pm0.1}$** | **$85.3_{\pm0.4}$** | **$67.4_{\pm0.8}$** | **$64.6_{\pm1.7}$** | **74.7** |
| SAGM | $85.3_{\pm2.5}$ | $80.9_{\pm1.1}$ | $97.1_{\pm0.4}$ | $77.8_{\pm0.5}$ | 85.3 | $78.9_{\pm1.2}$ | $79.8_{\pm1.0}$ | $44.7_{\pm1.8}$ | $55.6_{\pm1.1}$ | 64.8 |
| **UDIM w/ SAGM** | **$88.9_{\pm0.2}$** | **$86.2_{\pm0.3}$** | **$97.4_{\pm0.4}$** | **$79.5_{\pm0.8}$** | **88.0** | **$81.6_{\pm0.3}$** | **$84.8_{\pm1.2}$** | **$68.1_{\pm0.8}$** | **$63.3_{\pm0.9}$** | **74.5** |
| GAM | $85.5_{\pm0.6}$ | $81.1_{\pm1.0}$ | $96.4_{\pm0.2}$ | $81.0_{\pm1.7}$ | 86.0 | $79.1_{\pm1.3}$ | $79.7_{\pm0.9}$ | $46.3_{\pm0.6}$ | $56.6_{\pm1.1}$ | 65.4 |
| **UDIM w/ GAM** | **$87.1_{\pm0.9}$** | **$86.3_{\pm0.4}$** | **$97.2_{\pm0.1}$** | **$81.8_{\pm1.1}$** | **88.1** | **$82.4_{\pm0.9}$** | **$84.2_{\pm0.4}$** | **$68.8_{\pm0.8}$** | **$64.0_{\pm0.7}$** | **74.9** |

Table 5 and 7 shows the model performances for OfficeHome (Venkateswara et al., 2017) and DomainNet (Peng et al., 2019) dataset, respectively. Similar to the above tables, UDIM shows per-

Table 5: Results on OfficeHome dataset. $^*$ represents we got the results from Wang et al. (2023) and $'$ from Rame et al. (2022) considering Leave-One-Out Source Domain Generalization. For Single Source Domain Generalization case, we report the model performances generated under our experiment setting.

| Method | Leave-One-Out Source Domain Generalization | | | | | Single Source Domain Generalization | | | | |
| --- | --- | --- | --- | --- | --- | --- | --- | --- | --- | --- |
| | Art | Clipart | Product | Real World | Avg | Art | Clipart | Product | Real World | Avg |
| ERM | $61.4_{\pm1.0}$ | $53.5_{\pm0.2}$ | $75.9_{\pm0.2}$ | $77.1_{\pm0.2}$ | 67.0 | $55.6_{\pm0.6}$ | $52.8_{\pm1.6}$ | $50.3_{\pm1.1}$ | $59.4_{\pm0.3}$ | 54.5 |
| IRM$^*$ | $61.8^*_{\pm1.0}$ | $52.3^*_{\pm1.0}$ | $75.2^*_{\pm0.8}$ | $77.2^*_{\pm1.1}$ | $66.6^*$ | $54.9_{\pm0.3}$ | $53.2_{\pm0.6}$ | $48.6_{\pm0.7}$ | $59.2_{\pm0.1}$ | 54.0 |
| GroupDRO | $61.3_{\pm2.0}$ | $53.3_{\pm0.4}$ | $75.4_{\pm0.3}$ | $76.0_{\pm1.0}$ | 66.5 | $55.1_{\pm0.2}$ | $52.0_{\pm0.5}$ | $50.3_{\pm1.1}$ | $59.3_{\pm0.3}$ | 54.2 |
| OrgMixup | $63.7_{\pm1.1}$ | $55.4_{\pm0.1}$ | $77.1_{\pm0.2}$ | $78.9_{\pm0.6}$ | 68.8 | $56.0_{\pm1.1}$ | $54.4_{\pm1.3}$ | $50.4_{\pm0.4}$ | $61.0_{\pm0.7}$ | 55.5 |
| Mixup | $64.1_{\pm0.6}$ | $54.9_{\pm0.7}$ | $76.6_{\pm0.4}$ | $78.7_{\pm0.5}$ | 68.6 | $55.5_{\pm0.6}$ | $54.1_{\pm1.1}$ | $49.4_{\pm1.7}$ | $59.4_{\pm0.6}$ | 54.6 |
| CutMix | $63.2_{\pm0.3}$ | $52.1_{\pm1.7}$ | $77.2_{\pm0.8}$ | $78.1_{\pm0.6}$ | 67.7 | $53.5_{\pm0.8}$ | $52.2_{\pm1.2}$ | $47.7_{\pm1.7}$ | $60.2_{\pm0.4}$ | 53.4 |
| Mixstyle$^*$ | $51.1^*_{\pm0.3}$ | $53.2^*_{\pm0.4}$ | $68.2^*_{\pm0.7}$ | $69.2^*_{\pm0.6}$ | 60.4 | $44.3_{\pm0.5}$ | $29.8_{\pm1.2}$ | $33.6_{\pm0.5}$ | $48.5_{\pm0.9}$ | 39.0 |
| MTL | $60.1_{\pm0.5}$ | $52.0_{\pm0.3}$ | $75.7_{\pm0.3}$ | $77.2_{\pm0.4}$ | 66.3 | $55.3_{\pm0.3}$ | $53.3_{\pm0.4}$ | $49.0_{\pm0.3}$ | $60.4_{\pm0.1}$ | 54.5 |
| MLDG$^*$ | $63.7^*_{\pm0.3}$ | $54.5^*_{\pm0.6}$ | $75.9^*_{\pm0.4}$ | $78.6^*_{\pm0.1}$ | $68.2^*$ | - | - | - | - | - |
| MMD$^*$ | $63.0^*_{\pm0.1}$ | $53.7^*_{\pm0.9}$ | $76.1^*_{\pm0.3}$ | $78.1^*_{\pm0.5}$ | $67.7^*$ | $55.1_{\pm0.2}$ | $52.0_{\pm0.5}$ | $50.3_{\pm1.1}$ | $59.3_{\pm0.3}$ | 54.2 |
| CORAL | $64.1_{\pm0.5}$ | $54.5_{\pm1.7}$ | $76.2_{\pm0.4}$ | $77.8_{\pm0.5}$ | 68.2 | $55.6_{\pm0.6}$ | $52.8_{\pm1.6}$ | $50.3_{\pm1.1}$ | $59.4_{\pm0.3}$ | 54.5 |
| SagNet | $62.3_{\pm1.1}$ | $51.7_{\pm0.3}$ | $75.4_{\pm1.0}$ | $78.1_{\pm0.2}$ | 66.9 | $56.9_{\pm1.2}$ | $53.4_{\pm2.1}$ | $50.8_{\pm0.3}$ | $61.2_{\pm0.8}$ | 55.6 |
| ARM | $59.9_{\pm0.6}$ | $51.8_{\pm0.5}$ | $73.3_{\pm0.5}$ | $75.7_{\pm0.8}$ | 65.2 | $55.0_{\pm0.1}$ | $51.6_{\pm1.1}$ | $47.3_{\pm0.8}$ | $59.3_{\pm0.7}$ | 53.3 |
| DANN$^*$ | $59.9^*_{\pm1.3}$ | $53.0^*_{\pm0.3}$ | $73.6^*_{\pm0.7}$ | $76.9^*_{\pm0.5}$ | $65.9^*$ | $55.2_{\pm0.8}$ | $49.3_{\pm1.5}$ | $48.4_{\pm1.5}$ | $58.4_{\pm0.2}$ | 52.8 |
| CDANN$^*$ | $61.5^*_{\pm1.4}$ | $50.4^*_{\pm2.4}$ | $74.4^*_{\pm0.9}$ | $76.6^*_{\pm0.8}$ | $65.7^*$ | $55.2_{\pm0.7}$ | $49.9_{\pm1.4}$ | $47.6_{\pm1.3}$ | $58.6_{\pm0.5}$ | 52.8 |
| VREx | $61.4_{\pm1.0}$ | $52.2_{\pm0.2}$ | $76.1_{\pm0.6}$ | $77.6_{\pm0.9}$ | 66.8 | $55.5_{\pm0.6}$ | $52.6_{\pm0.2}$ | $49.1_{\pm1.0}$ | $59.3_{\pm0.6}$ | 54.1 |
| RSC$'$ | - | - | - | - | - | - | - | - | - | - |
| Fishr$'$ | $62.4_{\pm0.5}$ | $54.4_{\pm0.4}$ | $76.2_{\pm0.5}$ | $78.3_{\pm0.1}$ | 67.8 | $55.1_{\pm0.4}$ | $51.2_{\pm0.1}$ | $49.2_{\pm1.0}$ | $59.9_{\pm1.4}$ | 53.9 |
| RIDG | $63.6_{\pm0.7}$ | $55.0_{\pm0.9}$ | $76.0_{\pm0.8}$ | $77.5_{\pm0.7}$ | 68.0 | $56.8_{\pm0.5}$ | $55.4_{\pm0.7}$ | $50.5_{\pm0.3}$ | $60.9_{\pm0.1}$ | 55.9 |
| ITTA | $61.8_{\pm0.9}$ | $57.0_{\pm1.0}$ | $74.3_{\pm0.3}$ | $77.3_{\pm0.3}$ | 67.6 | $56.0_{\pm0.4}$ | $51.5_{\pm0.8}$ | $50.5_{\pm0.6}$ | $61.6_{\pm0.4}$ | 54.9 |
| SAM | $62.2_{\pm0.7}$ | $55.9_{\pm0.1}$ | $\mathbf{77.0}_{\pm0.9}$ | $78.8_{\pm0.6}$ | 68.5 | $56.9_{\pm0.4}$ | $53.8_{\pm1.1}$ | $50.9_{\pm0.7}$ | $61.5_{\pm0.8}$ | 55.8 |
| **UDIM** (w/ SAM) | $\mathbf{63.5}_{\pm1.3}$ | $\mathbf{58.6}_{\pm0.4}$ | $76.9_{\pm0.6}$ | $\mathbf{79.1}_{\pm0.3}$ | **69.5** | $\mathbf{58.1}_{\pm0.6}$ | $\mathbf{55.0}_{\pm0.9}$ | $\mathbf{53.8}_{\pm0.1}$ | $\mathbf{64.3}_{\pm0.2}$ | **57.8** |
| SAGM | $63.1_{\pm2.1}$ | $56.2_{\pm0.4}$ | $77.3_{\pm0.2}$ | $78.4_{\pm0.4}$ | 68.8 | $57.7_{\pm0.3}$ | $54.8_{\pm1.0}$ | $51.5_{\pm1.2}$ | $61.4_{\pm0.1}$ | 56.3 |
| **UDIM** (w/ SAGM) | $\mathbf{64.4}_{\pm0.3}$ | $\mathbf{57.3}_{\pm0.5}$ | $77.1_{\pm0.4}$ | $\mathbf{79.1}_{\pm0.3}$ | **69.5** | $\mathbf{58.5}_{\pm0.4}$ | $\mathbf{55.7}_{\pm0.6}$ | $\mathbf{54.5}_{\pm0.1}$ | $\mathbf{64.5}_{\pm0.4}$ | **58.3** |
| GAM | $64.0_{\pm0.5}$ | $58.6_{\pm1.2}$ | $77.5_{\pm0.1}$ | $79.3_{\pm0.2}$ | 69.8 | $59.4_{\pm0.6}$ | $56.1_{\pm0.9}$ | $53.3_{\pm0.6}$ | $63.4_{\pm0.2}$ | 58.1 |
| **UDIM** (w/ GAM) | $\mathbf{64.2}_{\pm0.3}$ | $57.4_{\pm0.9}$ | $77.5_{\pm0.1}$ | $79.3_{\pm0.3}$ | 69.6 | $58.7_{\pm0.3}$ | $55.7_{\pm0.0}$ | $53.6_{\pm0.4}$ | $\mathbf{64.4}_{\pm0.0}$ | 58.1 |

Table 6: Results on DomainNet datasets - Leave-One-Out (Multi) Source Domain Generalization

| Method | clipart | infograph | painting | quickdraw | real | sketch | Avg |
| --- | --- | --- | --- | --- | --- | --- | --- |
| ERM | $62.8_{\pm0.4}$ | $20.2_{\pm0.3}$ | $50.3_{\pm0.3}$ | $13.7_{\pm0.5}$ | $\mathbf{63.7}_{\pm0.2}$ | $52.1_{\pm0.5}$ | 43.8 |
| IRM | $48.5_{\pm2.8}$ | $15.0_{\pm1.5}$ | $38.3_{\pm4.3}$ | $10.9_{\pm0.5}$ | $48.2_{\pm5.2}$ | $42.3_{\pm3.1}$ | 33.9 |
| GroupDRO | $47.2_{\pm0.5}$ | $17.5_{\pm0.4}$ | $33.8_{\pm0.5}$ | $9.3_{\pm0.3}$ | $51.6_{\pm0.4}$ | $40.1_{\pm0.6}$ | 33.3 |
| MTL | $57.9_{\pm0.5}$ | $18.5_{\pm0.4}$ | $46.0_{\pm0.1}$ | $12.5_{\pm0.1}$ | $59.5_{\pm0.3}$ | $49.2_{\pm0.1}$ | 40.6 |
| MLDG | $59.1_{\pm0.2}$ | $19.1_{\pm0.3}$ | $45.8_{\pm0.7}$ | $13.4_{\pm0.3}$ | $59.6_{\pm0.2}$ | $50.2_{\pm0.4}$ | 41.2 |
| MMD | $32.1_{\pm13.3}$ | $11.0_{\pm4.6}$ | $26.8_{\pm11.3}$ | $8.7_{\pm2.1}$ | $32.7_{\pm13.8}$ | $28.9_{\pm11.9}$ | 23.4 |
| CORAL | $59.2_{\pm0.1}$ | $19.7_{\pm0.2}$ | $46.6_{\pm0.3}$ | $13.4_{\pm0.4}$ | $59.8_{\pm0.2}$ | $50.1_{\pm0.6}$ | 41.5 |
| SagNet | $57.7_{\pm0.3}$ | $19.0_{\pm0.2}$ | $45.3_{\pm0.3}$ | $12.7_{\pm0.5}$ | $58.1_{\pm0.5}$ | $48.8_{\pm0.2}$ | 40.3 |
| ARM | $49.7_{\pm0.3}$ | $16.3_{\pm0.5}$ | $40.9_{\pm1.1}$ | $9.4_{\pm0.1}$ | $53.4_{\pm0.4}$ | $43.5_{\pm0.4}$ | 35.5 |
| DANN$^*$ | $53.8^*_{\pm0.7}$ | $17.8^*_{\pm0.3}$ | $43.5^*_{\pm0.3}$ | $11.9^*_{\pm0.5}$ | $56.4^*_{\pm0.3}$ | $46.7^*_{\pm0.5}$ | $38.4^*$ |
| CDANN$^*$ | $53.4^*_{\pm0.4}$ | $18.3^*_{\pm0.7}$ | $44.8^*_{\pm0.3}$ | $12.9^*_{\pm0.2}$ | $57.5^*_{\pm0.4}$ | $46.7^*_{\pm0.2}$ | $38.9^*$ |
| VREx | $47.3_{\pm3.5}$ | $16.0_{\pm1.5}$ | $35.8_{\pm4.6}$ | $10.9_{\pm0.3}$ | $49.6_{\pm4.9}$ | $42.0_{\pm3.0}$ | 33.6 |
| RSC | $55.0_{\pm1.2}$ | $18.3_{\pm0.5}$ | $44.4_{\pm0.6}$ | $12.2_{\pm0.2}$ | $55.7_{\pm0.7}$ | $47.8_{\pm0.9}$ | 38.9 |
| Fishr$^*$ | $58.2^*_{\pm0.5}$ | $20.2^*_{\pm0.2}$ | $47.7^*_{\pm0.3}$ | $12.7^*_{\pm0.2}$ | $60.3^*_{\pm0.2}$ | $50.8^*_{\pm0.1}$ | $41.7^*$ |
| SAM | $\mathbf{64.5}_{\pm0.3}$ | $20.7_{\pm0.2}$ | $50.2_{\pm0.1}$ | $\mathbf{15.1}_{\pm0.3}$ | $62.6_{\pm0.2}$ | $52.7_{\pm0.3}$ | 44.3 |
| UDIM (w/ SAM) | $63.5_{\pm0.1}$ | $\mathbf{21.01}_{\pm0.1}$ | $\mathbf{50.63}_{\pm0.1}$ | $14.76_{\pm0.1}$ | $62.5_{\pm0.1}$ | $\mathbf{53.39}_{\pm0.1}$ | 44.3 |

formance improvement over sharpness-aware baselines, and good performances for each column consistently.

## D.5 ADDITIONAL ANALYSES

In this section, we additionally provide analysis on 1) Comparison with Shui et al. (2022), which utilzes both 1) data-based perturbation and 2) parameter-based regularization on their own framework. Afterwards, we provide visual inspections on the perturbed domain instances, which are constructed by Eq 11.

Table 7: Results on DomainNet datasets - Single Source Domain Generalization

| Method | clipart | infograph | painting | quickdraw | real | sketch | Avg |
|---|---|---|---|---|---|---|---|
| ERM | $27.5_{\pm0.5}$ | $26.6_{\pm0.1}$ | $28.5_{\pm0.3}$ | $7.1_{\pm0.1}$ | $28.9_{\pm0.4}$ | $29.4_{\pm0.3}$ | 24.7 |
| IRM | - | - | - | - | - | - | - |
| GroupDRO | $27.6_{\pm0.4}$ | $26.6_{\pm0.8}$ | $28.9_{\pm0.3}$ | $7.3_{\pm0.3}$ | $28.7_{\pm0.2}$ | $29.4_{\pm0.7}$ | 24.8 |
| OrgMixup | $28.3_{\pm0.1}$ | $\mathbf{27.3}_{\pm0.1}$ | $29.4_{\pm0.3}$ | $7.7_{\pm0.4}$ | $30.2_{\pm0.2}$ | $30.0_{\pm0.7}$ | 25.5 |
| Mixup | $27.5_{\pm0.3}$ | $26.9_{\pm0.8}$ | $28.6_{\pm0.4}$ | $7.0_{\pm0.1}$ | $28.7_{\pm0.2}$ | $29.6_{\pm0.6}$ | 24.7 |
| CutMix | $28.0_{\pm0.2}$ | $26.5_{\pm0.2}$ | $28.4_{\pm0.6}$ | $6.4_{\pm0.1}$ | $29.0_{\pm0.3}$ | $29.7_{\pm0.8}$ | 24.6 |
| Mixstyle | $18.3_{\pm0.8}$ | $14.5_{\pm0.3}$ | $19.6_{\pm0.2}$ | $5.0_{\pm0.1}$ | $20.5_{\pm1.3}$ | $21.3_{\pm0.8}$ | 16.5 |
| MTL | $27.3_{\pm0.2}$ | $26.6_{\pm0.3}$ | $28.7_{\pm0.1}$ | $7.8_{\pm0.1}$ | $28.2_{\pm0.2}$ | $28.7_{\pm0.6}$ | 24.5 |
| MLDG | - | - | - | - | - | - | - |
| MMD | $27.6_{\pm0.4}$ | $26.6_{\pm0.7}$ | $28.5_{\pm0.3}$ | $7.4_{\pm0.3}$ | $28.9_{\pm0.3}$ | $29.5_{\pm0.9}$ | 24.8 |
| CORAL | $18.1_{\pm12.8}$ | $17.4_{\pm12.3}$ | $18.9_{\pm13.4}$ | $4.8_{\pm3.4}$ | $19.5_{\pm13.8}$ | $20.2_{\pm14.3}$ | 16.5 |
| SagNet | $27.6_{\pm0.3}$ | $25.6_{\pm0.5}$ | $28.5_{\pm0.7}$ | $7.3_{\pm0.2}$ | $28.8_{\pm0.5}$ | $28.8_{\pm0.8}$ | 24.4 |
| ARM | $17.0_{\pm12.0}$ | $16.8_{\pm11.9}$ | $17.7_{\pm12.5}$ | $4.1_{\pm2.9}$ | $17.5_{\pm12.4}$ | $18.8_{\pm13.3}$ | 15.3 |
| DANN | $26.8_{\pm0.8}$ | $25.2_{\pm0.9}$ | $28.0_{\pm0.4}$ | $6.8_{\pm0.6}$ | $27.6_{\pm0.2}$ | $28.0_{\pm0.2}$ | 23.8 |
| CDANN | $26.8_{\pm0.7}$ | $25.8_{\pm0.2}$ | $27.6_{\pm0.2}$ | $6.8_{\pm0.6}$ | $27.6_{\pm0.2}$ | $28.0_{\pm0.2}$ | 23.8 |
| VREx | $18.6_{\pm13.1}$ | $17.3_{\pm12.3}$ | $19.3_{\pm13.6}$ | $4.7_{\pm3.3}$ | $19.3_{\pm13.7}$ | $19.9_{\pm14.1}$ | 16.5 |
| RSC | - | - | - | - | - | - | - |
| Fishr | $\mathbf{30.0}_{\pm0.3}$ | $26.6_{\pm0.7}$ | $28.9_{\pm0.3}$ | $7.5_{\pm0.7}$ | $28.4_{\pm0.7}$ | $28.9_{\pm0.3}$ | 24.7 |
| SAM | $28.4_{\pm0.2}$ | $26.9_{\pm0.1}$ | $29.1_{\pm0.4}$ | $6.9_{\pm0.5}$ | $30.0_{\pm0.2}$ | $29.8_{\pm0.7}$ | 25.2 |
| UDIM (w/ SAM) | $\mathbf{30.0}_{\pm0.1}$ | $23.8_{\pm0.4}$ | $\mathbf{31.0}_{\pm0.1}$ | $\mathbf{12.6}_{\pm0.1}$ | $\mathbf{30.7}_{\pm0.2}$ | $\mathbf{34.0}_{\pm0.3}$ | $\mathbf{27.0}$ |

### D.5.1 ANALYTICAL COMPARISON WITH SHUI ET AL. (2022)

UDIM and Shui et al. (2022) share a similar direction, as both methodologies involve 1) generating virtual samples through distinct data perturbation methods, and 2) implementing their own types of regularization on these samples.

Shui et al. (2022) introduced novel regularization techniques for the embedding function based on theoretical analysis. Their approach involves establishing an upper bound on the balanced error rate in the test environment, which is obtained from the combination of the balanced error rates in the source environments, the feature-conditional invariance, and the smoothness of the embedding function. In particular, they focused on minimizing the smoothness term by reducing the Frobenius norm of the Jacobian matrix with respect to the embedding function. To implement this regularization term over unobserved regions, they use virtual samples generated by a linear combination of samples from each source.

On the other hand, UDIM also introduces an upper bound on the generalization loss in an unknown domain. This bound includes the SAM loss on the source domain and a region-based loss disparity between the worst-case domain and the source domain. This disparity is implemented by a gradient variance-based objective outlined in Eq. 10. The objective involves a perturbed dataset constructed by perturbing samples, where the direction of perturbation is determined by the inconsistencies described in Eq. 7.

### D.5.2 ANALYTIC COMPARISON WITH DOMAIN AUGMENTATION AND ADVERSARIAL ATTACK

The proposed method, UDIM, involves applying arbitrary perturbation to a given instance to create a new instance, which is similar to domain augmentation and adversarial attacks. However, UDIM has the following differences and advantages compared to them.

**Comparison with domain augmentation** First, we taxonomize domain augmentation based on its dependency on learning signals such as parameters or loss, dividing it into not-learnable domain augmentation Zhang et al. (2018); Zhou et al. (2021); Li et al. (2021b) and learnable domain augmentation Zhang et al. (2018); Zhou et al. (2021); Li et al. (2021b). Please note that we briefly cite the representative methodologies among wide range of augmentation techniques.

While non-learnable domain augmentations are effective in generating new styles and may generalize well to specific types of domains, it does not guarantee generalization across wide range of

unseen domains, as discussed in the Theorem 3.2. In contrast, UDIM's data perturbation method is designed to generate perturbations towards the most vulnerable or worst domain from a parameter space perspective, enabling a reduction of the generalization bound in Eq 5, even in scenarios involving numerous unobserved domains. Additionally, it's important to note that these domain augmentation techniques could be applied orthogonally to the UDIM framework, for instance, by implementing domain augmentation prior to UDIM's data perturbation.

Learnable augmentations, similar to UDIM, determine its augmentation direction based on the current parameter response. However, these methodologies do not link their augmentation with a theoretical analysis to assure minimization of the target objective, which is left-hand side of Eq 5 of our manuscript. UDIM's data perturbation impacts the generalization bound from a parameter perspective as it takes into account a parameter loss curvature information, rather than just a single parameter point, when determining perturbations.

**Comparison with adversarial attack**   Adversarial attacks also introduce perturbations in the direction most vulnerable to the current parameters, but methodologies like FGSM Goodfellow et al. (2014) and PGD Madry et al. (2017) do not consider the local parameter curvature in their perturbation process. By integrating perturbations on instances with attention to parameter loss curvature; and parameter perturbation, we facilitate the modeling of inconsistency in unknown domains, as described in Eq 3. Having said that, Kim et al. (2023) also utilize worst-case instance selection on the active learning framework by utilizing the parameter perturbation. In the literature of coreset selection Feldman (2020), Shin et al. (2023) also utilizes the perturbed parameter region to attain samples which effectively represent whole dataset.

From a mathematical perspective, UDIM's data perturbation involves receiving not only gradients related to the simple cross-entropy loss but also additional gradients concerning the norm of gradient, as elaborated in Eq 7.

**Combination of domain augmentation with SAM optimizer**   We accordingly report the experimental results of models, which combines various domain augmentation techniques with SAM optimization in Table 8. We reported the average test accuracy for each domain in each setting. Applying SAM optimization to data instances of augmented domains led to mixed results: some methodologies improved, others didn't, but all still under-performed compared to UDIM. We hypothesize that the observed performance decline in certain augmentations combined with the SAM optimizer might stem from an unstable learning process. This instability may arise from attempting to minimize sharpness in the perturbed domain prematurely, before ensuring flatness in the source domain.

| Method | Leave-One-Out Source Domain Generalization | Single Source Domain Generalization |
|---|---|---|
| SAM | 85.9% | 64.8% |
| SAM w/ Mixup | 84.51% | 62.28% |
| SAM w/ Mixstyle | 86.56% | 68.59% |
| SAM w/ Simple Augment Li et al. (2021b) | 86.26% | 64.97% |
| SAM w/ advstyle Zhong et al. (2022) | 85.28% | 61.02% |
| UDIM | 88.7% | 74.7% |

Table 8: Performance comparison of models using combinations of domain augmentation variants and SAM optimizer, alongside UDIM model, on the PACS dataset.

### D.5.3   VISUAL INSPECTION ON THE PERTURBED DOMAIN

In this section, we aim to illustrate how perturbed instances are created depending on the size of $\rho$, which represents the magnitude of data perturbation. Each plot utilizes the PACS dataset and a model trained under the Single Source Domain Generalization setting.

Figure 6 displays instances perturbed through the original UDIM methodology. As the perturbation size increases, we can observe a gradual distortion of the image's semantics, highlighting the importance of selecting an appropriate perturbation size.

Figure 7 shows the scenario where the data perturbation of the original UDIM method is applied to the input channel, instead of input pixels. This approach of perturbing the channel over pixels has the advantage of maintaining the basic shape and line information of the image, ensuring the preservation of essential visual features.

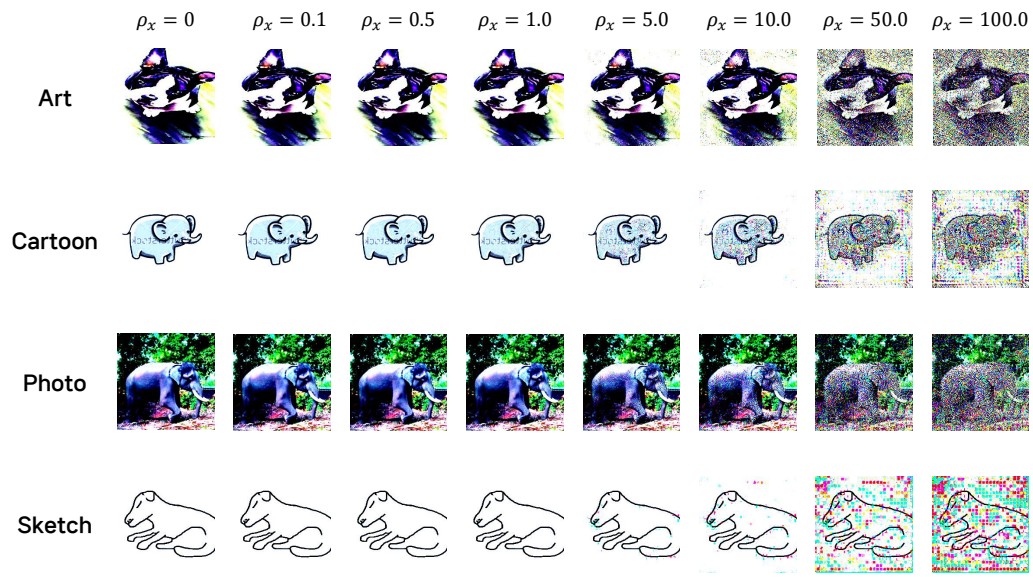

Figure 6: Pixel-perturbed domain instances by UDIM model trained on PACS dataset by varying the perturbation size

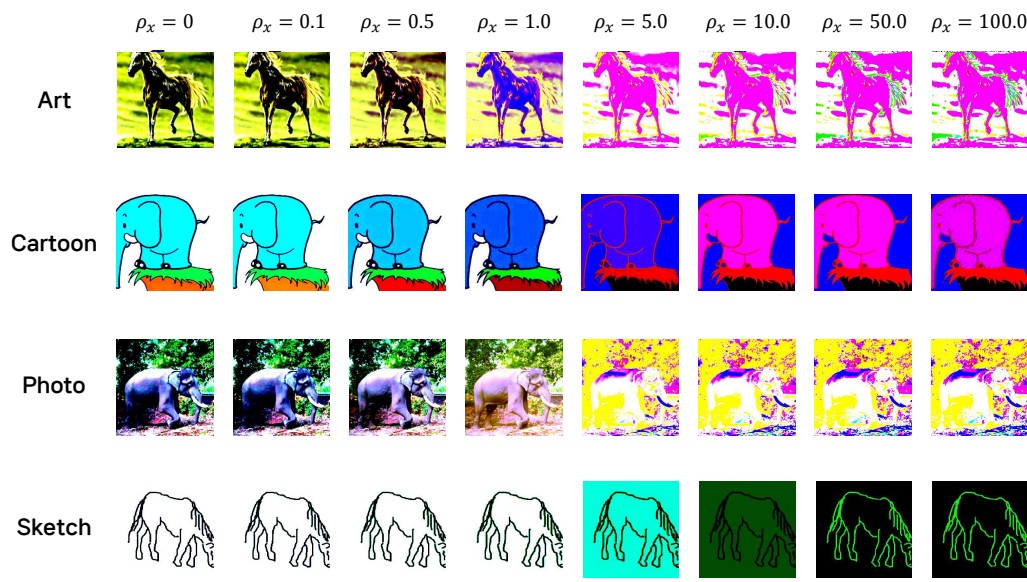

Figure 7: Channel-perturbed domain instances by UDIM model trained on PACS dataset by varying the perturbation size

