# OpenReview forum: "Unknown Domain Inconsistency Minimization for Domain Generalization"
_ICLR.cc/2024/Conference — ICLR 2024 poster_

### Official Review · Reviewer_ExGX · 2023-10-22

**Soundness:** 2 fair
**Presentation:** 3 good
**Contribution:** 2 fair
**Rating:** 6
**Confidence:** 5

**Summary:**

This paper highlights the importance of exploring the perturbation in data space to enhance the performance of domain generalization based on the sharpness-aware minimization approach. The authors proposed an unknown domain inconsistency minimization approach, which combines data perturbation to generate the worst-case unseen domains and weight perturbation which is sharpness-aware minimization. They further showed some theoretical analysis for their algorithm. Experiments have shown the effectiveness of the algorithm.

-----Post rebuttal

I appreciate the efforts in writing the response. While the efficiency problem is not solved, I acknowledge the efforts in trying to address it. Therefore, I increase my score to 6.

**Strengths:**

1. The idea of combining data perturbation with weight perturbation is interesting. The weight perturbation is a pretty “standard” approach to learn robust models and the combination of data perturbation can further enhance such robustness.
2. The paper is easy to understand and follow. The algorithm design is clear.
3. The experiments have shown the effectiveness of the approach.

**Weaknesses:**

1. The novelty of the approach is limited since the perturbation of input data is somewhat similar to existing approach called CrossGrad [Shankar et al., 2018] which also perturbs existing training data by reversing their gradient direction to generate unseen training domains. That being said, the paper can be seen as a combination of CrossGrad (by adding original data perturbation) and sharpness-aware minimization.
2. There lacks guaranteed proof showing that by perturbing the input to the size of $\rho$ (which is pretty similar to adversarial perturbation), the generated unseen domains can cover the entire space of unseen domains. It seems that with the size of $\rho$ becomes larger, the generalization risk (Eq. 3) is smaller. However, in that way, the sampling and generating efficiency will be heavily impacted. There are no clear analysis on the parameter $\rho$: what kind of perturbation can help the method already generalize the same as existing ones; better than existing ones, or worse than existing ones?
3. Insufficient experiments. The common benchmark in domain generalization is DomainBed, but the paper did not use it; instead, it used CIFAR-10-C and PACS, which are clearly not sufficient. DomainBed has other challenging datasets inclusing CMNIST and DomainNet, which should be tested.

**Questions:**

1. I cannot see the advantage of UDIM in Figure 5: it seems that the loss landscape (bottom right) of UDIM is no better than SAGM and GAM? Can authors offer further explanations?
2. On PACS, the results are not consistent with the original results from DomainBed. Authors should double check their results.
3. What is the efficiency of the method?
4. What do the generated domains look like?

---

> ### Author Response · Authors · 2023-11-18
> **Response to Reviewer ExGX [1/4]**
>
> We appreciate the reviewer for the constructive and thoughtful feedback. We answer for reviewer’s comments below.
>
> >**Q1. [Comparison with CrossGrad w/ SAM]** *(Weaknesses) The novelty of the approach is limited since the perturbation of input data is somewhat similar to existing approach called CrossGrad [Shankar et al., 2018] which also perturbs existing training data by reversing their gradient direction to generate unseen training domains. That being said, the paper can be seen as a combination of CrossGrad (by adding original data perturbation) and sharpness-aware minimization.*
>
> As reviewer ExGX noted, CrossGrad [1] shares a similar direction with UDIM's data perturbation in that it applies the reverse gradient of the loss objective as a perturbation. Having said that, UDIM and CrossGrad-based SAM optimization have the following differences.
>
> **In terms of data perturbation**
>
> * CrossGrad focuses solely on the reverse gradient of a particular loss, while UDIM's data perturbation considers the loss landscape in relation to the parameter space, as detailed in Eq (6) and (7). This unique perturbation enables the UDIM to align with Theorem 3.2 in the main paper.
>
> * Technically, CrossGrad applies perturbations to each instance via adversarial learning with a domain classifier. This process assumes the presence of multiple domains in the training dataset. This necessitates the use of multiple source domains during training, which could be a limitation in scenarios of the single source domain generalization. In contrast, UDIM, which provides perturbations in the direction most vulnerable from a parameter region perspective, operates effectively regardless of the number of domains in the training dataset.
>
> **In terms of parameter-based optimization**
>
> * Applying SAM optimization in conjunction with CrossGrad can be viewed as an effort to mitigate the loss sharpness in regions where instances are perturbed by CrossGrad. However, attempting to enforce flatness in the perturbed dataset without first regularizing the flatness in the source domain might not yield effective results. In contrast, UDIM's inconsistency minimization strategically minimizes the unknown domain inconsistency only within the flat regions of the source domain, where flat region is described as $N^{\gamma,\rho}_{s,\theta}$ in Eq (8) and (9). This approach effectively ensures that both the source domain and its associated perturbed domain maintain flat regions.
>
> **Performance comparison**
>
> * Having said that, we reproduced the CrossGrad and CrossGrad w/ SAM optimization for empirical comparison with UDIM. Technically, for the integration of CrossGrad and SAM optimization, SAM optimization wasn't applied in the segments related to data perturbation or domain classifier training. Rather, SAM optimization was confined to the training of the main classifier, which is \(\theta_{l}\). This training part corresponds to the 8-th line of Algorithm 1 in the CrossGrad paper.
>
> * As stated in the above paragraph, CrossGrad requires the availability of multiple source domains for experiment, so we only reproduced the results on Leave-One-Out Source Domain Generalization setting. Referring to the code available on GitHub (https://github.com/gpascualg/CrossGrad), it was initially developed for an MNIST implementation. Therefore, adapting it for a PACS dataset-based implementation was not a straightforward process. We replaced some network parts to suit the PACS dataset, and hyper-parameter tuning was also conducted as specified in the original paper.
>
> * Below table demonstrates that UDIM consistently demonstrates better performance compared to both CrossGrad and the CrossGrad-based SAM optimization. In the final manuscript, we will include an extensive comparison with CrossGrad and reflect the results of the corresponding experiments.
>
> | Method                    | PACS (Leave-One-Out Source Domain Generalization setting) |
> |---------------------------|-------------|
> | CrossGrad                 | 80.59%       |
> | CrossGrad w/ SAM Optimization | 78.54%   |
> | UDIM                      | 88.7%        |

---

> ### Author Response · Authors · 2023-11-18
> **Response to Reviewer ExGX [2/4]**
>
> >**Q2. [clear analysis on the parameter $\rho$]** *(Weaknesses) There lacks guaranteed proof showing that by perturbing the input to the size of \rho  (which is pretty similar to adversarial perturbation), the generated unseen domains can cover the entire space of unseen domains. It seems that with the size of \rho becomes larger, the generalization risk (Eq. 3) is smaller. However, in that way, the sampling and generating efficiency will be heavily impacted. There are no clear analysis on the parameter \rho: what kind of perturbation can help the method already generalize the same as existing ones; better than existing ones, or worse than existing ones?*
>
> Before answering the question, we would like to gently remind that in Theorem 3.2, the $\rho$ represents the perturbation with respect to the parameter $\theta$, while the perturbation on the input is denoted as $\rho_{x}$ in Eq (6) and Eq (7).
>
> As reviewer ExGX pointed out, both $\rho$, the perturbation with respect to the parameter $\theta$, and $\rho_{x}$, the perturbation of the input, can heavily impact the generalization performance of the trained model. We provide the discussion on each aspect as follows.
>
> **Analysis on $\rho_{x}$**
>
> * The size of $\rho_{x}$ represents the extent of exploration into unknown domains from a data space. However, since this exploration is implemented in the form of perturbations to each pixel of the input instance, too large perturbations can damage the semantics of the image instance. Therefore, the size of $\rho_{x}$ can be seen as determining the trade-off between preserving the semantics of the perturbed instance and exploring unknown domains.
>
> * To visually support in understanding the impact of the size on $\rho_{x}$, we have provided visualizations in the Figure 6 of Appendix D.4.2 in the revised manuscript. Figure 6 shows the domain-perturbed instances with varying $\rho_{x}$ sizes, offering a clear view of how different perturbation magnitudes affect the instances.
>
> * We can observe that if $\rho_{x}$ is excessively large, the semantics of the original image can be damaged. Also, in Figure 3 (a) of the revised manuscript, the ablation study of $\rho_{x}$ illustrates that an excessively large $\rho_{x}$ leads to the performance degradation. This empirically indicates that adjusting the size of $\rho_{x}$ involves a trade-off.
>
> * Additionally, in Eq (6) and (7), if the size of $\rho_{x}$ becomes too large, the approximation error by the first-order Taylor approximation may increase. To resolve this, an alternative optimization (e.g., constrained optimization) might be required instead of a first-order approximation. As the reviewer pointed out, this could lead to lower sampling efficiency. To ensure that efficiency issues do not hinder the real-world application of UDIM, UDIM utilizes an appropriately sized $\rho_{x}$ along with an efficient first-order Taylor approximation.
>
> **Analysis on $\rho$**
>
> If $\rho_{x}$ represents the extent of exploration in the data space from the input perspective, then $\rho$ can be understood as indicating the degree of exploration in the parameter space from the perspective of the parameter $\theta$. However, if $\rho$ is set excessively high, it can lead to following issues:
>
> * Similar to the previously mentioned $\rho_{x}$, setting a high $\rho$ can lead to a significant increase in estimation error during the Taylor approximation process. This issue is relevant in the context of both SAM variants and UDIM, as they both utilize Taylor approximation in parameter-perturbation-based optimization.
>
> * In the context of Theorem 3.2 in our manuscript, as the value of $\rho$ increases, it can excessively increase the terms on the right-hand side of the equation, except for the weight decay term. This could result in a looser upper bound in the right-hand side equation, potentially hindering the proper optimization of $\theta$.
>
> * [2] empirically demonstrates that as $\rho$ increases, it becomes more difficult to escape from saddle points.
>
> Based on the above reasons, SAM variants commonly utilize a value of $\rho$ = 0.05, and UDIM also follows this practice.

---

> ### Author Response · Authors · 2023-11-18
> **Response to Reviewer ExGX [3/4]**
>
> >**Q3. [Insufficient experiments]** *Insufficient experiments. The common benchmark in domain generalization is DomainBed, but the paper did not use it; instead, it used CIFAR-10-C and PACS, which are clearly not sufficient. DomainBed has other challenging datasets inclusing CMNIST and DomainNet, which should be tested.*
>
> We appreciate the reviewer ExGX's suggestion on experiments for additional datasets. Please refer the Appendix D.5, which demonstrates additional experimental results for OfficeHome and DomainNet, which are one of DomainBed datasets. UDIM similarly demonstrates improved performances in most cases compared to the SAM variants and other baselines.
>
> >**Q4. [Further explanations on Figure 5]** *I cannot see the advantage of UDIM in Figure 5: it seems that the loss landscape (bottom right) of UDIM is no better than SAGM and GAM? Can authors offer further explanations?*
>
> We claim that a favorable loss landscape for a specific dataset is characterized by 1) a low absolute value of loss, and simultaneously, 2) a flat loss landscape. This reasoning comes from the understanding that a flat landscape with a high loss value can signal under-fitting of $\theta$ to the dataset, while a sharply defined curvature with a very low loss might indicate over-fitting of $\theta$ to the dataset, both scenarios being critical to avoid in model optimization.
>
> Having said that, for a clearer understanding, we have added some quantitative values to Figure 5 in the revised manuscript. In this figure, we present both the max and min values to illustrate the absolute loss values. For the upper plots, which demonstrate the local sharpness of $\theta$, we also included the maximum eigenvalue of the Hessian matrix. This eigenvalue is commonly used as a measure to quantify the local sharpness of a given parameter $\theta$. It is stated that a lower maximum eigenvalue of the Hessian matrix indicates reduced local sharpness.
>
> Among the comparisons, UDIM not only shows the lowest loss values but also exhibits the smallest maximum eigenvalue, indicating the most favorable loss landscape as mentioned earlier. This underlines its robust ability to effectively generalize to the target domain.
>
> >**Q5. [Inconsistent results of PACS compared to DomainBed]** *On PACS, the results are not consistent with the original results from DomainBed. Authors should double check their results.*
>
> Thank you for the reviewer’s comments. As detailed in Appendix D.1 (Evaluation Detail) of our paper, we have reported the performance of each method based on the model selection, which is determined by the best accuracy time on the source validation dataset. In the original paper of DomainBed, this performance reporting is described as the *“model selection method using the training domain validation set.”* We also compared our reproduced performance with those results. In cases where the reported accuracy from DomainBed significantly surpassed ours, we substituted our reproduced performance with the reported performance from DomainBed. We indicate this change with an asterisk in the revised manuscript. It should be noted that the replaced performances are still lower than that of UDIM.
>
> >**Q6. [Efficiency of UDIM]** *What is the efficiency of the method?*
>
> UDIM requires additional gradient calculations during the optimization process, which adds computational burden. To minimize the burden, UDIM utilizes the backpack technique based on PyTorch, and for calculating the domain inconsistency as Eq (9), it employs gradients derived only from the classifier network's parameters, rather than the entire network.
>
> Having said that, we assessed each methodology's time complexity by measuring the training time required for 100 training iterations. As UDIM regularization is applied additively to the SAM variants, it requires additive computational cost to the SAM variants. The additional complexity is as follows.
>
> If the computation burden imposed by UDIM becomes significant for some environments, one way to reduce the computational cost is to apply UDIM regularization only at specific iterations, rather than at every iteration.
>
> | Method            | Time (sec/100 iterations) |
> |-------------------|---------------------------|
> | SAM               | 40                        |
> | SAM w/ UDIM       | 75                        |
> | SAGM              | 40                       |
> | SAGM w/ UDIM      | 86                       |
> | GAM               | 83                       |
> | GAM w/ UDIM       | 129                       |

---

> ### Author Response · Authors · 2023-11-18
> **Response to Reviewer ExGX [4/4]**
>
> >**Q7. [Visual Inspection of the perturbed domain instances]** *What do the generated domains look like?*
>
> Figure 6 of the Appendix D.4.2 in the revised manuscript visually demonstrates the perturbed instances based on the size of data perturbation, $\rho_{x}$. Perturbed images in the experiments effectively retain the semantics of their original classes at the utilized perturbation sizes of $\rho_{x}$.
>
> As part of an ablation study on instance perturbation, we also experimented with applying perturbations to the input channel, rather than the input pixels, to generate perturbed domain instances. Figure 7 in the revised manuscript provides the domain perturbed instances when perturbation was applied on the input channel. Changing the perturbation target to the input channel helps mitigate semantic distortion on the given image instance. Quantitatively, the UDIM performance with input channel perturbation remains similar to that of the original UDIM. Please refer the performances at the thread [Response to Reviewer KWQw [1/4]].
>
> **Reference**
>
> [1] Shankar, Shiv, et al. "Generalizing Across Domains via Cross-Gradient Training." International Conference on Learning Representations. 2018.
>
> [2] Kim, Hoki, et al. "Stability Analysis of Sharpness-Aware Minimization." arXiv preprint arXiv:2301.06308 (2023).

---

> > ### Comment · Reviewer_ExGX · 2023-11-20
> >
> > The reviewer appreciated the response. Most of the concerns are addressed now. However, w.r.t. efficiency, it seems unfixable: UDIM will double the training time, according to the response. Plus, authors also agreed that as $\rho$ increases, the sampling efficiency will be impacted. Together, training efficiency might be a significant issue especially in front of large-scale datasets. Considering that the advantages in accuracy are at cost of double training time, this approach lacks flexibility. That being said, the rating will not change.

---

> ### Author Response · Authors · 2023-11-21
> **Response to Reviewer ExGX**
>
> The authors appreciate Reviewer ExGX's feedback on the rebuttal response. As Reviewer ExGX pointed out, it's crucial to avoid significant increases in computational and training costs for the practical real-world applications.
>
> The methodology proposed in UDIM requires both data input perturbation and parameter perturbation. This dual approach is crucial for enabling the minimization of the upper bound of the target objective, as outlined in Theorem 3.2. Consequently, this requirement does lead to an additional computational cost.
>
> Having said that, the additional computational cost introduced by UDIM regularization is fixable. This can be achieved by applying UDIM optimization periodically and selectively on certain iterations, rather than on every iteration, in conjunction with SAM optimization.
>
> If UDIM regularization is applied every m iterations, the increase in training time becomes \(\frac{1}{m}\) of what would be caused by applying the original UDIM at every iteration. Also, if the performance of the model with UDIM regularization every m iterations does not significantly decline compared to the original UDIM regularization, then the benefits in accuracy would not entail doubling the training time.
>
> The table below presents the training times per 100 iterations for SAM and the original UDIM, as well as for models applying UDIM regularization every 2 iterations and every 4 iterations. For the case with 4 iterations, there is only about a 20% increase in training time compared to the original SAM.
>
> | Method           | Time (sec/100 iterations) |
> |-------------------|---------------------------|
> | SAM              | 40                       |
> | SAM w/ UDIM      | 75                       |
> | SAM w/ UDIM per 2 iterations   | 57.5                      |
> | SAM w/ UDIM per 4 iterations     | 48.75                     |
>
> Furthermore, the table below reports the accuracies of each model in the PACS-LOOSDG (Leave-One-Out Source Domain Generalization) and SSDG (Single Source Domain Generalization) settings. Given their experimental setup, the training times for both the LOOSDG and SSDG settings are identical. To reduce the noise from extra hyper-parameter tuning and focus exclusively on the impact of iteration period, the two new models (applying UDIM per 2 and 4 iterations) utilized the same hyper-parameters as the original UDIM, differing only in their iteration period. The reported accuracies correspond to the values in the 'Avg' column of Table 3 in the main paper, representing the average target domain accuracies across three different seeds.
>
>
> | Method                    | PACS (Leave-One-Out Source Domain Generalization) | PACS (Single Source Domain Generalization) |
> |----------------------------|-------------|----------|
> | SAM                       | 85.9% | 64.8% |
> | SAM w/ UDIM         | 88.7% | 74.7%    |
> | SAM w/ UDIM per 2 iterations     | 88.3%       | 74.7%   |
> | SAM w/ UDIM per 4 iterations     | 88.0%         | 74.5%   |
>
>
> **The performance results of above table provide us two findings.**
>
> * First, increasing the iteration interval for UDIM regularization slightly reduces performance. This indicates that UDIM regularization has a positive impact when integrated with SAM optimization.
> * Second, even with a 4-iteration interval, the performance decline compared to the original UDIM is not significant, and we can observe a substantial performance improvement over the original SAM optimization.
>
> This suggests that significant improvements can be achieved with UDIM, without bearing the burden of doubling the training time.

---

> > ### Author Response · Authors · 2023-11-23
> > **Dear Reviewer ExGX**
> >
> > The authors greatly appreciate your time and effort in reviewing this submission, and eagerly await your updated response. The discussion deadline is approaching, and we have only a few hours left.
> >
> > Please help us to review our responses once again and kindly let us know whether they fully or partially address your concerns and if our explanations are in the right direction.
> >
> > Best Regards,
> >
> > The authors of Submission 3600

---

> > > ### Comment · Reviewer_ExGX · 2023-11-23
> > >
> > > I thank the authors for providing this insightful table. I understand that there is no need to apply the algorithm in each iteration, but this just leads to another problem: there is an extra hyperparameter (which iteration to apply to?) to tune, which actually did not solve the problem. But I acknowledge such trial and encourage the authors to include such discussion in the revision and further think about reducing the computing cost. I increased my score to 6. No further responses are needed.

---

> > > > ### Author Response · Authors · 2023-11-23
> > > >
> > > > We sincerely appreciate the reviewer's dedication in reviewing our responses and the comments on the time-complexity issues. Your comments are of great value to us, and the authors will take them into account carefully to enhance the quality of our paper.

---

### Official Review · Reviewer_DtQi · 2023-10-23

**Soundness:** 3 good
**Presentation:** 3 good
**Contribution:** 3 good
**Rating:** 5
**Confidence:** 3

**Summary:**

This paper, titled "Unknown Domain Inconsistency Minimization for Domain Generalization," introduces an approach to improve domain generalization using Unknown Domain Inconsistency Minimization (UDIM) in combination with Sharpness-Aware Minimization (SAM) variants.  The paper's novelty lies in its approach to improving domain generalization by focusing on both parameter and data perturbed regions. UDIM is introduced as a novel concept, addressing the need for robust generalization to unknown domains, which is a critical issue in domain generalization. The idea of perturbing the instances in the source domain dataset to emulate unknown domains and aligning flat minima across domains is innovative. While SAM-based approaches have shown promise in DG, UDIM extends the concept to address specific shortcomings, which is a novel contribution.

**Strengths:**

1. The paper appears to be technically sound. It provides a well-defined problem statement for domain generalization and formulates UDIM as an optimization objective. The authors validate the theoretical foundation of UDIM and provide empirical results across various benchmark datasets, demonstrating its effectiveness. The methodology is explained clearly, and the experiments are well-documented.

2. The paper is well-structured and clearly written. It provides a thorough introduction, problem definition, and a detailed explanation of the proposed method. The methodology is presented step by step, and mathematical notations are used effectively. The experimental setup and results are also presented in a clear and organized manner. However, the paper is quite technical, and readers with less familiarity with domain generalization and machine learning might find some sections challenging to follow.


3. The paper addresses an important challenge in domain generalization, namely, the ability of models to generalize to unknown domains. The proposed UDIM method appears to be effective in improving the performance of existing SAM-based approaches, as demonstrated through experimental results. The potential impact of this paper on the AI research community is significant, particularly in the field of domain generalization.

**Weaknesses:**

1. Baselines. The baselines are not enough because the latest DG method is Fisher, which is published at 2022.

**Questions:**

see weakness

---

> ### Author Response · Authors · 2023-11-18
> **Response to Reviewer DtQi [1/1]**
>
> We appreciate the reviewer for the constructive and thoughtful feedback. We answer for reviewer’s comments below.
>
> >**Q1. [Further requirement on the latest baselines]** *(Weaknesses) Baselines. The baselines are not enough because the latest DG method is Fisher, which is published at 2022.*
>
> We appreciate the reviewer’s valuable comments. We have incorporated the latest published baselines in domain generalization, which are [1] and [2], into our experimental results. Please see Table 3,4 and 5 of the revised manuscript.
>
> From the original paper of [1] and [2], each method only reports the experimental performance on the Leave-One-Out Source Domain Generalization setting. Therefore, in the case of the LOOSDG (Leave-One-Out Source Domain Generalization) setting, we have included the reported performances of each baseline in our table. Conversely, for the SSDG (Single Source Domain Generalization) setting, we independently reproduced and reported the performances. It should be noted that the reported performance of new baselines in the LOOSDG setting is slightly lower compared to each of our original baselines. We are in the process of reproducing the LOOSDG performance for each baseline, and we will sequentially update our results as they are completed. For SSDG setting, the results of [1] and [2] show improved performance compared to the original baselines, yet they still fall short of the performances from UDIM. In the final version, we will include and report results for more diverse datasets.
>
> We also would like to gently remind the reviewer DtQi that, although the SAM variants methodologies are orthogonally applied to the UDIM framework, the SAGM [3] and GAM [4] utilized in our experiments were also published in 2023.
>
> During the rebuttal process, we have conducted performance comparisons with arbitrary baselines combining SAM-based optimization with various data augmentations or input perturbations. For more details, please refer to the thread [Response to Reviewer KWQw [1/4]]. These experimental results illustrate that UDIM consistently delivers improved performance compared to various combinations of data perturbation and parameter perturbation.
>
> **Reference**
>
> [1] Chen, Liang, et al. "Domain generalization via rationale invariance." Proceedings of the IEEE/CVF International Conference on Computer Vision. 2023.
>
> [2] Chen, L., Zhang, Y., Song, Y., Shan, Y., & Liu, L. (2023). Improved Test-Time Adaptation for Domain Generalization. In Proceedings of the IEEE/CVF Conference on Computer Vision and Pattern Recognition (pp. 24172-24182).
>
> [3] Wang, P., Zhang, Z., Lei, Z., & Zhang, L. (2023). Sharpness-aware gradient matching for domain generalization. In Proceedings of the IEEE/CVF Conference on Computer Vision and Pattern Recognition (pp. 3769-3778).
>
> [4] Zhang, Xingxuan, et al. "Gradient norm aware minimization seeks first-order flatness and improves generalization." Proceedings of the IEEE/CVF Conference on Computer Vision and Pattern Recognition. 2023.

---

> > ### Comment · Reviewer_DtQi · 2023-11-22
> > **Official Comment**
> >
> > 1. The performance of Fisher on the PACS dataset in this paper differs significantly from the results reported in [1] (85.5 vs. 81.3). It raises the question: what factors contribute to this discrepancy?
> >
> > 2. Furthermore, the newly introduced baselines exhibit inferior performance compared to ERM across most benchmarks. This raises concerns about their utility or suggests that these baselines may not offer significant value.
> >
> > 3. The decision to exclusively employ the leave-one-out setting warrants clarification. Understanding the rationale behind this choice would contribute to a more comprehensive evaluation.
> >
> > [1] Chen, Liang, et al. "Domain generalization via rationale invariance." Proceedings of the IEEE/CVF International Conference on Computer Vision. 2023.

---

> ### Author Response · Authors · 2023-11-22
> **Clarifying the misunderstanding on the PACS result from new baselines (RIDG [1] and ITTA [2])**
>
> We would like to sincerely apologize our mistakes in the reported results for the Leave-One-Out Source Domain Generalization setting of the PACS dataset in Table 3.
>
>
> Given the limited rebuttal time window, we directly reproduced the RIDG [1] and ITTA [2] for Single Source Domain Generalization setting. However, for the Leave-One-Out Source Domain setting of RIDG and ITTA, we cited the results from their original papers [1,2], and this reference to reported results is noted in the caption of Table 3.
>
>
> Meanwhile, as the reviewer pointed out their inferior performances, we discovered that the reported performances were originated from a simpler backbone vision network (ResNet-18) while we utilize ResNet-50 to generate the results in Table 3 except RIDG and ITTA. The situation also applies to the Table 4 of OfficeHome dataset.
>
>
> To construct a fair comparison, we are running experiments of RIDG and ITTA with ResNet-50, which will be finished within 7 hours. We will report the results within the rebuttal time window, and we will let the reviewer DtQi know when it is finished and the updated manuscript is uploaded.

---

> ### Author Response · Authors · 2023-11-22
> **Updated response to Reviewer DtQi [1/2]**
>
> ## Response to additional questions
>
> **(Answer to Q1)** The performance difference between [1] and UDIM is from the different deployment of backbone network. 81.3 is from Fishr with ResNet-18 ([1]) and 85.5 is from Fishr with ResNet-50 (Our paper).
>
>  **(Answer to Q2)** The inferior performance of [1] and [2] compared to ERM is from our mistake. During the rebuttal period, we mistakenly reported the ResNet-18 based results of [1] and [2] in Tables 3 and 4 of our paper, which are reported based on ResNet-50 experiments.
>
> **(Answer to Q3)** Our updated manuscript presents the results of our own implementation, where we have reproduced the performances for both Single Source Domain Generalization and Leave-One-Out Source Domain Generalization settings, specifically for new baselines [1] and [2]. Please note that the updated manuscript contains the reproduced performances of two baselines, without exclusive employment of the leave-one-out setting.
>
> We sincerely apologize for our mistakes on the performance reports. For a correct understanding on our experiments, we provide further detailed response as follows:
>
> ***
>
> ## Revised manuscript with updated results on [1] and [2]
>
> We would like to announce the upload of the revised manuscript, which now includes the reproduced results for RIDG [1] and ITTA [2] under the Leave-One-Out Source Domain Generalization setting for the PACS and OfficeHome datasets. These updated results are presented in Table 4 and Table 5, respectively. Our manuscript has already undergone a first revision during the rebuttal period. For clarity, we have marked the changes made in the first revision in red, and those made in the second revision in blue.
>
> As previously mentioned in our discussion, the reported performances in the original papers for each baseline were based on the ResNet-18 backbone. For a fair comparison with the performances in our tables, which were obtained utilizing the ResNet-50, we have reproduced the final reported performances of [1] and [2] using the ResNet-50 backbone network.
>
> Although it is provided in the revised manuscript, for ease of visibility, we present below the tables showing the newly reproduced performance of [1] and [2] under the Leave-One-Out Source Domain Generalization scenario, along with the performances of UDIM.
>
> The RIDG and ITTA models each demonstrate improved performances compared to ERM. However, UDIM still consistently exhibits superior performances.
>
> * **PACS results based on ResNet-50 (Leave-One-Out Source Domain Generalization setting)**
>
> | Model               | Art | Cartoon | Photo | Sketch | Avg |
> |---------------------|------------|------------|------------|------------|------------|
> | ERM                 | 86.9$\pm$2.3 | 79.5$\pm$1.5 | 96.6$\pm$0.5 | 78.2$\pm$4.1 | 85.3       |
> | RIDG            | 86.3$\pm$1.1 | 81.0$\pm$1.0 | 97.4$\pm$0.7 | 77.5$\pm$2.5 | 85.5 |
> | ITTA            | 87.9$\pm$1.4 | 78.6$\pm$2.7 | 96.2$\pm$0.2 | 80.7$\pm$2.2 | 85.8 |
> | **UDIM** w/ SAM     | **88.5$\pm$0.1** | **86.1$\pm$0.1** | **97.3$\pm$0.1** | **82.7$\pm$0.1** | **88.7** |
>
> * **OfficeHome results based on ResNet-50 (Leave-One-Out Source Domain Generalization setting)**
>
> | Model               | Art | Clipart | Product | Real World | Avg |
> |---------------------|------------|------------|------------|------------|------------|
> | ERM                 | 61.4$\pm$1.0 | 53.5$\pm$0.2 | 75.9$\pm$0.2 | 77.1$\pm$0.2 | 67.0       |
> | RIDG            | 63.6$\pm$0.7 | 55.0$\pm$0.9 | 76.0$\pm$0.8 | 77.5$\pm$0.7 | 68.0 |
> | ITTA            | 61.8$\pm$0.9 | 57.0$\pm$1.0 | 74.3$\pm$0.3 | 77.3$\pm$0.3 | 67.6 |
> | **UDIM** w/ SAM     | **63.5$\pm$1.3** | **58.6$\pm$0.4** | **76.9$\pm$0.6** | **79.1$\pm$0.3** | **69.5** |
>
> ***
>
> ## Credibility of the reproduced results on [1] and [2]
>
> While we have reproduced the performance as mentioned, there is a need to validate whether the reproduction on ResNet-50 was conducted accurately. To address this, we also reproduced the performances of [1] and [2] with a ResNet-18 backbone based on our code, and checked whether these performances significantly deviates from the reported results in the original paper. The table below demonstrates that the reported performances of [1] and [2] do not significantly differ from our own implementations.
>
> * **PACS results of [1] and [2] based on ResNet-18 (Leave-One-Out Source Domain Generalization setting)**
>
> | Model              | Art | Cartoon | Photo | Sketch | Avg |
> |---------------------|------------|------------|------------|------------|------------|
> | RIDG  (reported in [1])  | 82.4$\pm$1.0 | 76.7$\pm$0.6 | 95.3$\pm$0.1 | 76.7$\pm$0.3 | 82.8 |
> | RIDG  (reproduced) | 83.8$\pm$0.47 | 75.5$\pm$1.2 | 95.5$\pm$0.87 | 73.99$\pm$2.2 | 82.2 |
> | ITTA  (reported in [2]) | 84.7$\pm$0.4 | 78.0$\pm$0.4 | 94.5$\pm$0.4 | 78.2$\pm$0.3 | 83.8 |
> | ITTA  (reproduced) | 83.26$\pm$1.5 | 75.9$\pm$0.4 | 95.6$\pm$0.4 | 78.5$\pm$1.41 | 83.3 |

---

> > ### Author Response · Authors · 2023-11-22
> > **Updated response to Reviewer DtQi [2/2]**
> >
> > We sincerely apologize for causing the confusion to the reviewer DtQi. It is from incorrect performance reports of our new baselines, [1] and [2], which were mistakenly presented during the rebuttal period. We also sincerely thank the reviewer DtQi for carefully reviewing our paper and pointing out such issues. Having said that, we respectfully request a reconsideration of the scoring of our paper based on our reproduced results and the updated manuscript. In the final version, we will report performances on more diverse range of baselines and various datasets.

---

> > > ### Author Response · Authors · 2023-11-23
> > > **Dear Reviewer DtQi**
> > >
> > > The authors greatly appreciate your time and effort in reviewing this submission, and eagerly await your response. We understand you might be quite busy. However, the discussion deadline is approaching, and we have only a few hours left.
> > >
> > > We have provided detailed responses to every one of your concerns/questions. Please help us to review our responses once again and kindly let us know whether they fully or partially address your concerns and if our explanations are in the right direction.
> > >
> > > Best regards,
> > >
> > > The authors of Submission 3600

---

### Official Review · Reviewer_x9yc · 2023-10-26

**Soundness:** 2 fair
**Presentation:** 2 fair
**Contribution:** 2 fair
**Rating:** 6
**Confidence:** 4

**Summary:**

This paper considered both parameter- and data-  perturbation in domain generalization. The method is inspired by sharpness aware minimization (SAM). A theoretical analysis is further conducted to show the importance of different perturbations. Finally, the model is deployed in standard benchmarks with improved performance.

------Post-rebuttal
I would appreciate the rebuttal, which addressed my concerns. I would maintain a positive rating.

**Strengths:**

1. This paper considered a reasonable solution in domain generalization. Both parameter and data perturbations are conducted for a robust OOD generalization.
2. The idea seems novel for me in some settings.
3. Extensive empirical results.

Based on these points, I would recommend a borderline positive.

**Weaknesses:**

1. Sometimes I find it a bit hard to understand the rationale of the proposed approach. Why do we need to consider both parameter and data perturbation? For example, in paper [1], a theoretical analysis is proposed, which is analogous to equation (11) as the parameter robust.
2. Does the choice of data perturbation matter? We know we may face many different possible data-augmentation approaches. Which method(s) do you think should work in this scenario?
3. Is it possible to consider the subgroup distribution shift in the context of fairness such as papers [2-3]? A short discussion could be great.

References:

[1] On the Benefits of Representation Regularization in Invariance based Domain Generalization. Machine Learning Journal (MLJ) 2022.

[2] On Learning Fairness and Accuracy on Multiple Subgroups. Neural Information Processing Systems (NeurIPS) 2022.

[3] Fair Representation Learning through Implicit Path Alignment. International Conference on Machine Learning (ICML) 2022.

**Questions:**

See weakness part.

**Details Of Ethics Concerns:**

It could be good to discuss how the proposed method to encourage fairness in machine learning.

---

> ### Author Response · Authors · 2023-11-18
> **Response to Reviewer x9yc [1/2]**
>
> We appreciate the reviewer for the constructive and thoughtful feedback. We answer for reviewer’s comments below. Please refer the citation at the end of the thread.
>
> >**Q1. [Need of both parameter and data perturbation and comparison with [1]]** *(Weaknesses) Sometimes I find it a bit hard to understand the rationale of the proposed approach. Why do we need to consider both parameter and data perturbation? For example, in paper [1], a theoretical analysis is proposed, which is analogous to equation (11) as the parameter robust.*
>
> **Why do we need both parameter and data perturbation for UDIM?**
>
> UDIM states, through Theorem 3.2 in the manuscript, that perturbations based on parameter space combined with explorations in the data space can reduce the upper bound of the target objective in domain generalization. The right-hand side of Eq (5) represents the upper bound. The first term on the right-hand side corresponds to parameter perturbation through SAM-based optimization. The second term is the proposed inconsistency term of UDIM. As discussed in Section 3.3, the tractable implementation of this term inevitably necessitates both data and parameter perturbation. Therefore, the translation of Theorem 3.2’s right-hand side into a implementable form indispensably requires the concurrent application of both parameter and data perturbation.
>
> **Comparison with [1]**
>
> As reviewer x9yc noted, UDIM and [1] share a similar direction, as both methodologies involve 1) generating virtual samples through distinct data perturbation methods, and 2) implementing their own types of regularization on these samples. In the Appendix D.4.1 of the revised manuscript, we discuss these methodological similarities by citing [1] to illustrate the comparison.
>
> >**Q2. [About the choice of data perturbation]** *(Weaknesses) Does the choice of data perturbation matter? We know we may face many different possible data-augmentation approaches. Which method(s) do you think should work in this scenario?*
>
> During the training phase, if the target domain is known, it's possible to find an optimal data augmentation or perturbation tailored to it, and the most effective augmentation will likely differ for each target domain. However, domain generalization operates under the assumption that the target domain is unknown, making it challenging to identify the optimal augmentation strategy.
>
> Given the difficulty in identifying the optimal augmentation, UDIM utilizes basic pixel-based data perturbation to create perturbed domain instances. However, by shifting the target of data perturbation from input pixels to input channels, we can also implement a different style of perturbation. It should be noted that the loss and direction of perturbation remain the same; only the target of perturbation changes to the input channel.
>
> Qualitatively, Figure 6 and 7 in the revised manuscript presents visualizations of perturbed instances corresponding to changes in perturbation size, showcasing both versions: one applied to UDIM's input pixels and the other to UDIM's input channels. Each figure shows that each perturbation carries a distinct style of the perturbed domain.
>
> Quantitatively, the table in the thread of [Response to Reviewer KWQw [1/4]] reports the performance of original version of UDIM and the new version of UDIM, which is marked as *UDIM w/ input channel perturbation*, where perturbation is applied to the input channel. Additionally, we also report the experimental results of models combining various domain augmentation methodologies with SAM optimization. By applying SAM optimization to data instances of the augmented domains, some methodologies show improved performance, while others do not exhibit such improvements. Also these methods still shows lower results compared to the UDIM performance. UDIM w/ input channel perturbation provides different styles of perturbation, yet their performance does not show significant differences compared to the original UDIM.

---

> ### Author Response · Authors · 2023-11-18
> **Response to Reviewer x9yc [2/2]**
>
> >**Q3. [Connection to the subgroup distribution shift in the context of fairness]** *(Weaknesses) Is it possible to consider the subgroup distribution shift in the context of fairness such as papers [2-3]? A short discussion could be great.*
>
> We sincerely appreciate the reviewer x9yc's valuable suggestion. As reviewer x9yc noted, [2] and [3] focus on fairness across multiple subgroups. These studies are related to UDIM in their objective to improve the performance of a single classifier applied simultaneously across various groups. A key distinction is that in [2] and [3], training samples from multiple groups are available for training the classifier, albeit in limited numbers. In contrast, our work seeks to enhance accuracies across multiple target domains that remain unseen during the training phase.
>
> Given this context, to enhance fairness across multiple groups, it becomes necessary to identify specific domains or groups where the current model parameter underperforms. Since UDIM's data perturbation considers the most vulnerable direction from the parameter space perspective, it can assist in identifying these groups or creating representations for unidentified groups.
>
> Specifically, by increasing the number of original group instances through UDIM's data perturbation, it is possible to further alleviate the limited sample number issue that [2] addresses and attempts to resolve. In the final version of our manuscript, we will delve deeper into the discussion of subgroup distribution shifts from a fairness perspective. During this process, we will appropriately cite [2], [3], and other related works in conjunction.
>
> **Reference**
>
> [1] On the Benefits of Representation Regularization in Invariance based Domain Generalization. Machine Learning Journal (MLJ) 2022.
>
> [2] On Learning Fairness and Accuracy on Multiple Subgroups. Neural Information Processing Systems (NeurIPS) 2022.
>
> [3] Fair Representation Learning through Implicit Path Alignment. International Conference on Machine Learning (ICML) 2022.

---

> > ### Author Response · Authors · 2023-11-23
> > **Dear Reviewer x9yc**
> >
> > The authors greatly appreciate your time and effort in reviewing this submission, and eagerly await your response. We understand you might be quite busy. However, the discussion deadline is approaching, and we have only a few hours left.
> >
> > We have provided detailed responses to every one of your concerns/questions. Please help us to review our responses once again and kindly let us know whether they fully or partially address your concerns and if our explanations are in the right direction.
> >
> > Best Regards,
> > The authors of Submission 3600

---

### Official Review · Reviewer_KWQw · 2023-11-01

**Soundness:** 3 good
**Presentation:** 3 good
**Contribution:** 2 fair
**Rating:** 6
**Confidence:** 3

**Summary:**

This paper focuses on domain generalization through extending the flattened loss landscape in the perturbed parameter space to the perturbed data space. Specifically, they first simulate the unknown domains via perturbing the source data, and then reduce the loss landscape inconsistency between source domains and the perturbed domains, thereby achieving robust generalization ability for the unobserved domain. Theoretical analysis and extensive experiments demonstrate the effectiveness and superiority of this method.

**Strengths:**

1.This work extends the parameter perturbation in existing SAM optimization to data perturbation, achieving loss landscape alignment between source domains and unknown domains. Experiments show the validity of the proposed objective.

2.This work establishes an upper bound for DG by merging SAM optimization with the proposed objective.

3.The proposed objective can be combined with multiple SAM optimizers and further enhance their performance, demonstrating the necessity of the loss landscape consistency between source and unknown domains.

**Weaknesses:**

1.I believe that the proposed data perturbation method is consistent in both ideology and essence with traditional domain augmentation and adversarial attack techniques. So, what is the main difference and advantage of the proposed objective? And what if combining some domain augmentation techniques with the SAM optimizers?

2.How to guarantee that the perturbed data is still meaningful, rather than generating some noisy samples? If so, will enforced the loss landscape alignment across domains bring negative impact? Besides, the unknown distribution may not necessarily be within the scope of perturbed data region.

3.What is the sampling strategy for sampling data from $D_s$ to generate perturbed data?

4.Is the optimization only conducted on the perturbed samples during the formal training phase? Would it be more stable to train on both the source and perturbed samples simultaneously?

5.Since the training procedure involves gradient calculation, what is the time complexity after applying the BackPACK technique?

**Questions:**

See weakness.

---

> ### Author Response · Authors · 2023-11-18
> **Response to Reviewer KWQw [1/4]**
>
> We appreciate the reviewer for the constructive and thoughtful feedback. We answer for reviewer’s comments below. Please refer the citation at the end of the thread.
>
> >**Q1. [Comparison with domain augmentation and adversarial attack]** *(Weaknesses) I believe that the proposed data perturbation method is consistent in both ideology and essence with traditional domain augmentation and adversarial attack techniques. So, what is the main difference and advantage of the proposed objective? And what if combining some domain augmentation techniques with the SAM optimizers?*
>
> As reviewer KWQw has noted, the proposed method, UDIM, involves applying arbitrary perturbation to a given instance to create a new instance, which is similar to domain augmentation and adversarial attacks. However, UDIM has the following differences and advantages compared to them.
>
> **Comparison with domain augmentation**
>
> - First, we taxonomize domain augmentation based on its dependency on learning signals such as parameters or loss, dividing it into not-learnable domain augmentation [1,2,3] and learnable domain augmentation [4,5]. Please note that we briefly cite the representative methodologies among wide range of augmentation techniques.
>
> - While non-learnable domain augmentations are effective in generating new styles and may generalize well to specific types of domains, it does not guarantee generalization across wide range of unseen domains, as discussed in the main paper's Theorem 3.2. In contrast, UDIM’s data perturbation method is designed to generate perturbations towards the most vulnerable or worst domain from a parameter space perspective, enabling a reduction of the generalization bound in Eq (5), even in scenarios involving numerous unobserved domains. Additionally, it's important to note that these domain augmentation techniques could be applied orthogonally to the UDIM framework, for instance, by implementing domain augmentation prior to UDIM's data perturbation.
>
> - Learnable augmentations, similar to UDIM, determine its augmentation direction based on the current parameter response. However, these methodologies do not link their augmentation with a theoretical analysis to assure minimization of the target objective, which is left-hand side of Eq (5) of our manuscript. UDIM's data perturbation impacts the generalization bound from a parameter perspective as it takes into account a parameter loss curvature information, rather than just a single parameter point, when determining perturbations.
>
> **Comparison with adversarial attack**
>
> - Adversarial attacks also introduce perturbations in the direction most vulnerable to the current parameters, but methodologies like FGSM [6] and PGD [7] do not consider the local parameter curvature in their perturbation process. By integrating perturbations on instances with attention to parameter loss curvature; and parameter perturbation, we facilitate the modeling of inconsistency in unknown domains, as described in Eq (3).
>
> - From a mathematical perspective, UDIM's data perturbation involves receiving not only gradients related to the simple cross-entropy loss but also additional gradients concerning the norm of gradient, as elaborated in Eq (7).
>
> - Due to the similarities with adversarial attacks, we have already conducted an ablation study where we replaced UDIM's data perturbation part to the adversarial attack techniques. Please refer the center of Figure 4 in the manuscript. UDIM with the original perturbation demonstrates improved performance compared to UDIM with adversarial attack-based perturbations.
>
> **Combination of domain augmentation techniques with SAM optimizer**
>
> - We accordingly report the experimental results of models, which combines various domain augmentation techniques with SAM optimization as follows. We reported the average test accuracy for each domain in each setting. Applying SAM optimization to data instances of augmented domains led to mixed results: some methodologies improved, others didn't, but all still under-performed compared to UDIM.
>
> - We hypothesize that the observed performance decline in certain augmentations combined with the SAM optimizer might stem from an unstable learning process. This instability may arise from attempting to minimize sharpness in the perturbed domain prematurely, before ensuring flatness in the source domain.
>
>
>
> | Method                     | PACS (Leave-One-Out Source Domain Generalization) | PACS (Single Source Domain Generalization)  |
> |----------------------------|-------------|----------|
> | SAM                        | 85.9% | 64.8%  |
> | SAM w/ Mixup [1]           | 84.51%        | 62.28%     |
> | SAM w/ Mixstyle [2]        | 86.56%        | 68.59%    |
> | SAM w/ Simple Augment [3]  |  86.26%        | 64.97%     |
> | SAM w/ advstyle [5]        | 85.28%       | 61.02%    |
> | UDIM                       | 88.7%        | 74.7%     |
> | UDIM w/ input channel perturbation                     | 88.62%        | 74.52%     |

---

> ### Author Response · Authors · 2023-11-18
> **Response to Reviewer KWQw [2/4]**
>
> >**Q2. [Validity of perturbed domain instances]** *(Weaknesses) How to guarantee that the perturbed data is still meaningful, rather than generating some noisy samples? If so, will enforced the loss landscape alignment across domains bring negative impact? Besides, the unknown distribution may not necessarily be within the scope of perturbed data region.*
>
> As reviewer KWQw noted, applying perturbations beyond a certain threshold could compromise the semantics of the image. Ultimately, matching the loss curvature with such semantically damaged images could potentially have a negative effect on the learning process. However, if the perturbation size is set too small, it may fail to sufficiently cover the target unknown distribution, leading to insufficient data space exploration. In that sense, **the size of \(\rho_{x}\) would determine the trade-off between maintaining the semantics of the data instance and exploring the unknown domain distribution.**
>
> Figure 6 in the revised manuscript visually demonstrates the perturbed instances based on the different size of data perturbation, $\rho_{x}$. Perturbed images in the experiments effectively retain the semantics of their original classes at the utilized perturbation sizes.
>
> Having said that, UDIM perturbs image pixels directly, and too much perturbation can distort image semantics. To address this, we also tried the perturbations targeting the input channel instead of input pixels, allowing for larger perturbations while better preserving semantics. Figure 7 in the revised manuscript provides the domain perturbed instances when perturbation was applied on the input channel. Changing the perturbation target in this manner helps mitigate semantic distortion. For quantitative comparison, UDIM w/ input channel perturbation in the above table still shows similar results to original UDIM, which utilizes pixel perturbation.
>
> >**Q3. [Sampling strategy]** *(Weaknesses) What is the sampling strategy for sampling data from D_{s} to generate perturbed data?*
>
> In terms of sampling, we follow the experimental setting of [8]. For the Leave-One-Out Source Domain Generalization setting, where multiple source domains are utilized for training, we randomly draw an equal number of samples from each domain in every iteration. For instance, in a PACS dataset setting where P, A, and C are the source domains and S is the target, we extract an equal number of B samples from P, A, and C in each iteration. Consequently, the whole mini-batch size for that iteration becomes 3*B.  In the case of Single Source Domain Generalization, since only a single source is utilized, i.i.d mini-batch sampling is performed in each iteration.

---

> ### Author Response · Authors · 2023-11-18
> **Response to Reviewer KWQw [3/4]**
>
> >**Q4. [Specification of training for each phase]** *(Weaknesses) Is the optimization only conducted on the perturbed samples during the formal training phase? Would it be more stable to train on both the source and perturbed samples simultaneously?*
>
> Before answering the question, the term 'formal training' could be interpreted either as the official training phase where UDIM regularization is applied, or as the 'former warm-up training' prior to the application of UDIM regularization. To provide clear information to the reviewer, we recap the precise training method for each phase.
>
> In the warm-up stage, or the former phase of training, our approach begins with conducting SAM optimization on the raw source domain dataset, denoted as $D_{s}$. This phase corresponds to the first for-loop of Algorithm 1 in Appendix C. This step is crucial for flattening the loss landscape associated with the source domain. It's important to note that during this phase, we do not employ any perturbed instances.
>
> In the latter stage, where UDIM is applied, we simultaneously utilize both the source domain instances, represented as $D_{B}$, and the perturbed domain instances, denoted as $\tilde{D}_{B}$, to match the loss landscape between these two datasets. This phase corresponds to the second for-loop of Algorithm 1.
>
> >**Q5. [Time complexity of UDIM]** *(Weaknesses) Since the training procedure involves gradient calculation, what is the time complexity after applying the BackPACK technique?*
>
> As the reviewer commented, UDIM requires additional gradient calculations during the optimization process, which adds computational burden. To minimize the burden, UDIM utilizes the backpack technique based on PyTorch, and for calculating the domain inconsistency as per Eq (9), it employs gradients derived only from the classifier network's parameters, rather than the entire network.
>
> We assessed each methodology's time complexity by measuring the training time required for 100 training iterations. As UDIM regularization is applied additively to the SAM variants, it requires additive computational cost to the SAM variants. The additional complexity is as follows.
>
> UDIM necessitates both data input perturbation and parameter perturbation. Consequently, this requirement does lead to an additional computational cost.  If the computation burden imposed by UDIM becomes significant, one way to reduce the computational cost is to apply UDIM regularization only at specific iterations, rather than at every iteration. For instance, if UDIM regularization is applied only once every four iterations, the computational cost attributed to UDIM would be reduced to a quarter over the entire training iterations.
>
> We are currently conducting experiments where UDIM regularization is applied only at certain intervals (for example, every 2 or 4 iterations), rather than at every iteration. ~~As soon as the results of these experiments are available, we intend to share them with Reviewer KWQw, within the rebuttal period.~~ **Please refer the last thread, which provides additional analyses on time complexity of UDIM.**
>
> | Method            | Time (sec/100 iterations) |
> |-------------------|---------------------------|
> | SAM               | 40                        |
> | SAM w/ UDIM       | 75                        |
> | SAGM              | 40                       |
> | SAGM w/ UDIM      | 86                       |
> | GAM               | 83                       |
> | GAM w/ UDIM       | 129                   |

---

> ### Author Response · Authors · 2023-11-18
> **Response to Reviewer KWQw [4/4]**
>
> **Reference**
>
> [1] Zhang, Hongyi, et al. "mixup: Beyond Empirical Risk Minimization." International Conference on Learning Representations. 2018.
>
> [2] Zhou, Kaiyang, et al. "Domain Generalization with MixStyle." International Conference on Learning Representations. 2020.
>
> [3] Li, Pan, et al. "A simple feature augmentation for domain generalization." Proceedings of the IEEE/CVF International Conference on Computer Vision. 2021.
>
> [4] Volpi, Riccardo, et al. "Generalizing to unseen domains via adversarial data augmentation." Advances in neural information processing systems 31 (2018).
>
> [5] Zhong, Zhun, et al. "Adversarial style augmentation for domain generalized urban-scene segmentation." Advances in Neural Information Processing Systems 35 (2022): 338-350.
>
> [6] Goodfellow, I. J., Shlens, J., & Szegedy, C. (2015). Explaining and Harnessing Adversarial Examples. In Y. Bengio & Y. LeCun (Eds.), 3rd International Conference on Learning Representations, ICLR 2015, San Diego, CA, USA, May 7-9, 2015, Conference Track Proceedings.
>
> [7] Madry, A., Makelov, A., Schmidt, L., Tsipras, D., & Vladu, A. (2018). Towards Deep Learning Models Resistant to Adversarial Attacks. International Conference on Learning Representations.
>
> [8] Wang, P., Zhang, Z., Lei, Z., & Zhang, L. (2023). Sharpness-aware gradient matching for domain generalization. In Proceedings of the IEEE/CVF Conference on Computer Vision and Pattern Recognition (pp. 3769-3778).

---

> ### Author Response · Authors · 2023-11-21
> **Updated response to Reviewer KWQw (Additional response for Q5)**
>
> **This is additional response for Q5 [Time complexity of UDIM]**
>
> The methodology proposed in UDIM requires both data input perturbation and parameter perturbation. This dual approach is crucial for enabling the minimization of the upper bound of the target objective, as outlined in Theorem 3.2. Consequently, this requirement does lead to an additional computational cost.
>
> Having said that, the additional computational cost introduced by UDIM regularization is fixable. This can be achieved by applying UDIM optimization periodically and selectively on certain iterations, rather than on every iteration, in conjunction with SAM optimization.
>
> If UDIM regularization is applied every m iterations, the increase in training time becomes \(\frac{1}{m}\) of what would be caused by applying the original UDIM at every iteration. Also, if the performance of the model with UDIM regularization every m iterations does not significantly decline compared to the original UDIM regularization, then the benefits in accuracy would not entail doubling the training time.
>
> The table below presents the training times per 100 iterations for SAM and the original UDIM, as well as for models applying UDIM regularization every 2 iterations and every 4 iterations. For the case with 4 iterations, there is only about a 20% increase in training time compared to the original SAM.
>
> | Method           | Time (sec/100 iterations) |
> |-------------------|---------------------------|
> | SAM              | 40                       |
> | SAM w/ UDIM      | 75                       |
> | SAM w/ UDIM per 2 iterations   | 57.5                      |
> | SAM w/ UDIM per 4 iterations     | 48.75                     |
>
> Furthermore, the table below reports the accuracies of each model in the PACS-LOOSDG (Leave-One-Out Source Domain Generalization) and SSDG (Single Source Domain Generalization) settings. Given their experimental setup, the training times for both the LOOSDG and SSDG settings are identical. To reduce the noise from extra hyper-parameter tuning and focus exclusively on the impact of iteration period, the two new models (applying UDIM per 2 and 4 iterations) utilized the same hyper-parameters as the original UDIM, differing only in their iteration period. The reported accuracies correspond to the values in the 'Avg' column of Table 3 in the main paper, representing the average target domain accuracies across three different seeds.
>
>
> | Method                    | PACS (Leave-One-Out Source Domain Generalization) | PACS (Single Source Domain Generalization) |
> |----------------------------|-------------|----------|
> | SAM                       | 85.9% | 64.8% |
> | SAM w/ UDIM         | 88.7% | 74.7%    |
> | SAM w/ UDIM per 2 iterations     | 88.3%       | 74.7%   |
> | SAM w/ UDIM per 4 iterations     | 88.0%         | 74.5%   |
>
>
> **The performance results of above table provide us two findings.**
>
> * First, increasing the iteration interval for UDIM regularization slightly reduces performance. This indicates that UDIM regularization has a positive impact when integrated with SAM optimization.
> * Second, even with a 4-iteration interval, the performance decline compared to the original UDIM is not significant, and we can observe a substantial performance improvement over the original SAM optimization.
>
> This suggests that significant improvements can be achieved with UDIM, without bearing the burden of doubling the training time.

---

> > ### Author Response · Authors · 2023-11-23
> > **Dear Reviewer KWQw**
> >
> > The authors greatly appreciate your time and effort in reviewing this submission, and eagerly await your response. We understand you might be quite busy. However, the discussion deadline is approaching, and we have only a few hours left.
> >
> > We have provided detailed responses to every one of your concerns/questions. Please help us to review our responses once again and kindly let us know whether they fully or partially address your concerns and if our explanations are in the right direction.
> >
> > Best Regards,
> >
> > The authors of Submission 3600

---

### Author Response · Authors · 2023-11-22
**Global response to the reviewers**

Dear Reviewers,
We would like to express our gratitude for the constructive comments you provided on our work. We genuinely appreciate the time and effort you have dedicated to this process. We uploaded the updated manuscript with our responses to address your comments. The revised or added parts of the manuscript are colored with **red** (for first revision) or **blue** (for second revision).

If you require additional information or have any further inquiries about our paper, please let us know. We remain open to discussions and are ready to provide any necessary clarifications until the rebuttal response deadline.

---

### Meta-Review · Area_Chair_nYXG · 2023-12-07

**Metareview:**

This paper proposes a training method to improve domain generalization.  It builds upon  Sharpness-Aware Minimization (SAM), which seeks to find model parameters theta such that the worst-case loss in a small neighborhood of theta is  minimal (flat minima). The new idea is to create a surrogate worst-case unseen domain by perturbing training examples of the source domain, and to minimize, besides SAM loss, the worst-case difference between losses in the two domains.  The reviewers find the idea interesting and the theoretical and empirical work reasonable.  The overall support for the paper is not very strong because the “technical contributions are modest”. In addition, the work can be strengthened by further experiments with real-world benchmarks such as WILDS (https://github.com/facebookresearch/DomainBed).

**Justification For Why Not Higher Score:**

No reviewer is excited about this work.

**Justification For Why Not Lower Score:**

Reasonable work.

---

### Decision · Program_Chairs · 2024-01-16

Accept (poster)